# Trajectory of immune evasion and cancer progression in hepatocellular carcinoma

Phuong H. D. Nguyen [1,18], Martin Wasser[1,2,18], Chong Teik Tan[3], Chun Jye Lim[1], Hannah L. H. Lai[4], Justine Jia Wen Seow[4], Ramanuj DasGupta [4], Cheryl Z. J. Phua[4], Siming Ma[4], Jicheng Yang[2], Sheena D/O Suthen[1], Wai Leong Tam [4,5,6,7], Tony K. H. Lim[2,8], Joe Yeong[2,8,9], Wei Qiang Leow[2,8], Yin Huei Pang[10], Gwyneth Soon[10], Tracy Jiezhen Loh[8], Wei Keat Wan[8], Chung Yip Chan[2,11], Peng Chung Cheow[2,11], Han Chong Toh[2,12], Alfred Kow[13], Yock Young Dan[14], Juinn Huar Kam[2,11], Shridhar Iyer [13], Krishnakumar Madhavan[13], Alexander Chung[2,11], Glenn K. Bonney[13], Brian K. P. Goh[2,11], Naiyang Fu [2], Victor C. Yu [3], Weiwei Zhai[4,15,16], Salvatore Albani [1,2✉], Pierce K. H. Chow [11,17✉] & Valerie Chew [1,2✉]

Immune evasion is key to cancer initiation and later at metastasis, but its dynamics at intermediate stages, where potential therapeutic interventions could be applied, is undefined. Here we show, using multi-dimensional analyses of resected tumours, their adjacent non-tumour tissues and peripheral blood, that extensive immune remodelling takes place in patients with stage I to III hepatocellular carcinoma (HCC). We demonstrate the depletion of anti-tumoural immune subsets and accumulation of immunosuppressive or exhausted subsets along with reduced tumour infiltration of CD8 T cells peaking at stage II tumours. Corresponding transcriptomic modification occur in the genes related to antigen presentation, immune responses, and chemotaxis. The progressive immune evasion is validated in a murine model of HCC. Our results show evidence of ongoing tumour-immune co-evolution during HCC progression and offer insights into potential interventions to reverse, prevent or limit the progression of the disease.

[1] Translational Immunology Institute (TII), SingHealth-DukeNUS Academic Medical Centre, Singapore 169856, Singapore. [2] Duke-Nus Medical School, Singapore 169857, Singapore. [3] Department of Pharmacy, National University of Singapore, Singapore 117559, Singapore. [4] Genome Institute of Singapore (GIS), Agency for Science, Technology and Research (A*STAR), Singapore 138672, Singapore. [5] School of Biological Sciences, Nanyang Technological University, Singapore 637551, Singapore. [6] Cancer Science Institute of Singapore, National University of Singapore, Singapore 117599, Singapore. [7] Department of Biochemistry, Yong Loo Lin School of Medicine, National University of Singapore, Singapore 117596, Singapore. [8] Department of Anatomical Pathology, Singapore General Hospital, Singapore 169856, Singapore. [9] Institute of Molecular and Cell Biology (IMCB), Agency for Science, Technology and Research (A*STAR), Singapore 138673, Singapore. [10] Department of Pathology, National University Hospital, Singapore 119074, Singapore. [11] Department of Hepatopancreatobiliary and Transplant Surgery, Division of Surgery and Surgical Oncology, Singapore General Hospital and National Cancer Centre Singapore, Singapore 169608, Singapore. [12] Division of Medical Oncology, National Cancer Centre Singapore, Singapore 169610, Singapore. [13] Division of Hepatobiliary & Pancreatic Surgery, Department of Surgery, University Surgical Cluster, National University Health System, Singapore 119074, Singapore. [14] Department of Medicine, Yong Loo Lin School of Medicine, National University of Singapore, Singapore 117597, Singapore. [15] Key Laboratory of Zoological Systematics and Evolution, Institute of Zoology, Chinese Academy of Sciences, Beijing 100107, China. [16] Center for Excellence in Animal Evolution and Genetics, Chinese Academy of Sciences, Kunming, Yunan 650223, China. [17] Academic Clinical Programme for Surgery, SingHealth Duke-NUS Academic Medical Centre (AMC), Singapore 169857, Singapore. [18]These authors contributed equally: Phuong H.D. Nguyen, Martin Wasser. ✉email: salvo@duke-nus.edu.sg; pierce.chow@duke-nus.edu.sg; valerie.chew@duke-nus.edu.sg

epatocellular carcinoma (HCC) remains a critical cancer burden ranking as the fourth most common cancer-related mortality worldwide[1]. The immune micro-environment plays a critical role in tumour development, particularly in HCC, where long-standing liver inflammation or chronic hepatitis infection often precedes carcinogenesis[2–4]. Moreover, the immune composition of the tumour micro-environment (TME) contributes to disease prognosis[5–9], evidencing its importance in determining clinical outcomes of HCC patients. Indeed, tumour initiation and progression require escape from the immune system, representing an ongoing arms race between cancer and host. Studies have defined immune evasion as an early event in lung cancer[10,11] or late event during metastasis in advanced colorectal cancer[12]. Thus both early and late phases of immune evasion could exist, with different characteristics and mechanisms to pave the way towards tumour progression. However, what happens in between these points throughout the intermediate stages of cancer is unknown.

By exploring the changing immune landscapes in patients with TNM (tumour size, lymph node involvement and metastasis) stage I–III HCC spanning across tumours, adjacent non-tumour tissues and blood, we show early and sustained immune escape. A second wave of immune evasion and concurrent tumour evolution peaks at stage II HCC. Our findings demonstrate continual immune-tumour co-evolution throughout tumour progression and identify intermediate tumour stage as a potential key interventional point for immunotherapy to prevent widespread HCC.

## Results

**Immune landscapes modification during HCC progression.** To define the immune modifications along tumour progression, we first characterised the local and systemic immune landscapes in 38 treatment-naïve patients with TNM stage (S) I–III (S1-S3) HCC ($n = 16$ S1, $n = 12$ S2 and $n = 10$ S3), who underwent resection as first-line therapy (Supplementary Table 1). We collected two to five regions per tumour (T) samples (total $T = 136$) to account for intratumoural heterogeneity reported previously[13,14], along with 38 matched adjacent non-tumour liver tissues (N) and peripheral blood (P) samples. All patients were clinically comparable, except for those parameters related to disease stage (i.e. tumour size and microvascular invasion) (Supplementary Table 1).

We first examined the phenotypes of immune cells in T, N and P compartments using mass cytometry by time-of-flight (CyTOF) to measure protein expression of 33 immune markers on or within individual single cells (Supplementary Table 2). Using Phenograph clustering[15] and our in-house built web-based CyTOF analytics pipeline[16], we identified 36 immune cell clusters (C0-C35) (Fig. 1a and Supplementary Fig. 1a) that were assigned to different immune subsets and major immune lineages according to their relative immune marker expressions (Fig. 1b and Supplementary Fig. 1b). UMAP dimensionality reduction plots showed the presence of more distinct immune clusters in T and to a lesser extent in N and P across all stages that were not confounded by patient-level differences (Supplementary Fig. 1c). Indeed, increased phenotypic heterogeneity or immune complexity was significant in tumours compared to adjacent non-tumour liver microenvironment or peripheral blood (Supplementary Fig. 1d).

From all three tumour stages, we identified the same 13 clusters that showed significant differences in their frequencies across tissue types: P, N and T using one-way ANOVA test (Fig. 1c and Supplementary Table 3). Among these, we observed PD-1+GBloCD45RO+CD8+ memory T (C1), CD69+CD8+ memory T cells (C9) and PD-L1+CD45RO+CD4+ memory T cells (C13) were enriched in both N and T compared to P (Supplementary

Fig. 2a); while the antigen-presenting HLA-DR+CD19+ B cells (C8) and HLA-DR+CD14+ myeloid cells (C10), as well as CD27+CD45RO−CD4+ naïve T cells (C17) were significantly reduced in both N and T compared to P (Supplementary Fig. 2b). This shows a general accumulation of memory subsets and depletion of antigen-presenting cells in the N and T compartments as compared to P from all three tumour stages. More importantly, we observed enrichment of potentially exhausted and suppressive immune subsets in tumours: Foxp3+CD152+TIGIT+CD4+ regulatory T cells (Treg) (C4), PD1+CD103+CD45RO+CD8+ resident memory T cells (TRM) (C7) and PD1+ CD45RO+CD4+ memory T cells (C19) (Fig. 1d). Conversely, the frequencies of immunoactive subsets, including GB+CD56+ NK cells (C2) and GB+CD56+CD8+ NKT cells (C18) were significantly lower in T than N or P (Supplementary Fig. 2b).

Taken together, these data indicate that immune evasion is established early and maintained throughout the following stages of tumour development.

**Progressive immune evasion peaking at stage II HCC.** Next, to test if the immune landscapes within the same microenvironment differed among tumour stages, we compared the frequencies of the immune cell clusters among the three tumour stages within each tissue compartment using one-way ANOVA analysis (Supplementary Table 4). In P, Treg (C4), CD4+CD27+CD45RO− naïve T cells (C17) and CD3+Ki67+ proliferating T cells (C20) were less abundant in S2 compared to S1 or S3 (Fig. 2a, b). Whereas in N, increased CD8+PD-1+CD103+CD45RO+ TRM (C7) coupled with reduced CD3+CD56+ natural killer (NK) T cells (C15) were observed in S2 (Fig. 2a, c).

Tumours tissues exhibited the greatest inter-stage heterogeneity: six clusters showed significant difference in abundance between stages (Supplementary Table 4), particularly at S2 tumours (Fig. 2a). S2 tumours exhibited lower levels of pro-inflammatory or activated immune subsets: including GB+CD16+CD56+ active NK cells (C2, $p = 0.0471$, S2 vs S3), PD-1−GB+CD45RO+ (C6, $p = 0.0220$, S1 vs S2), and PD-1−CD69+CD45RO+ (C9, $p = 0.009$, S1 vs S2; $p = 0.0184$, S2 vs S3) active memory CD8+ T cells (Fig. 2d). Conversely, the frequencies of potentially exhausted and immunosuppressive clusters: PD-1+GBloCD45RO+CD8+ exhausted T cells (C1, $p = 0.0458$, S2 vs S3), Treg (C4, $p = 0.0188$, S1 vs S2), and PD-1+GBloCD45RO+CD4+ exhausted memory T cells (C19, $p = 0.0073$, S2 vs S3) were significantly higher in S2 tumours than in either S1 or S3 (Fig. 2d).

We also validated the above data using another clustering method, FlowSOM[17] and found consistent trend of immune changes along tumour progression. Particularly, the immuno-suppressive Treg and exhausted CD8+ T cells peaked in S2 tumours, while the active CD8+ T cells and active NK cells showed the lowest frequencies in S2 tumours (Supplementary Fig. 3a, b and Supplementary Table 5). Likewise, we validated the depletion of Treg in S2 PBMC and reduction of NKT cluster in S2 NILs (Supplementary Fig. 3c and Supplementary Table 5).

Manual gating (Supplementary Fig. 4a) validated that PD-1+GB−CD8+ exhausted T cells, Foxp3+CD152+CD4+ Treg, and PD-1+CD45RO+CD4+ memory T cells were enriched; while GB+CD56+ active NK cells and PD-1−GB+CD8+ activated T cells were depleted, specifically in S2 tumours compared to S1 or S3 tumours or both (Fig. 2e). Of note, the PD-1+CD8+ T cells expressed low level of granzyme B (Median $21.85\% \pm 23.05\%$) and co-expressed high level of CTLA-4 (Median $66.2\% \pm 21.71\%$), Tim3 (Median $= 72.4\% \pm 17.61\%$) and to a lesser extent Lag3 (Median $36.3\% \pm 17.07\%$) (Supplementary Fig. 4b–d), supporting their exhausted

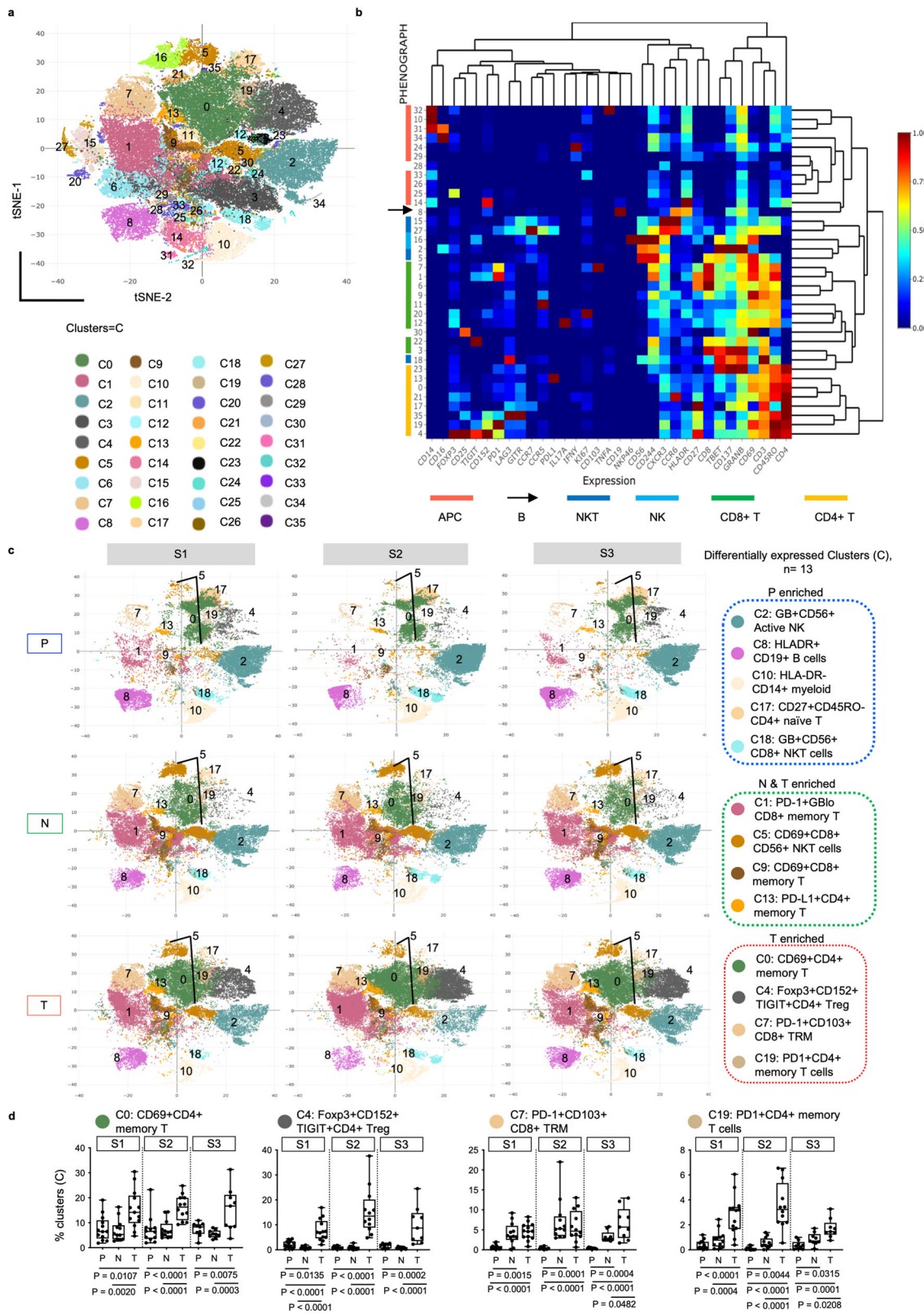

phenotypes as also shown by our previous study of exhausted PD-1+CD8+ T cells in HCC[18]. These data indicate the apparent peak of immune evasion in S2 HCC tumours, followed by a potential partial immune recovery at S3.

**Single-cell immune trajectory along tumour progression.** Next, we performed single-cell RNA sequencing on the tumour-

infiltrating immune cells as previously described[19] and analyzed their pseudotime trajectory along tumour progression using Monocle, an algorithm to learn the sequence of gene expression changes each cell must go through as part of a dynamic biological process[20]. Setting S1 tumours as the starting point, we first explored the T cells trajectory along tumour progression S1-S3. Consistently, we observed a more distinct immune phenotype at

**Fig. 1 Diverse immune landscapes of HCC along tumour progression. a** Global tSNE plot showing 36 Phenograph immune clusters (C) from tumours (T), adjacent non-tumour liver tissues (N) and peripheral blood (P). Each cluster represented by one colour. **b** Heatmap representation of the same 36 immune clusters (rows) with normalised protein expression of 33 markers (columns) from all samples. The colour bars and arrow on the left indicate the major immune cell lineages shown below. **c** tSNE plots showing distributions and frequencies of 13 differentially expressed immune clusters across P, N or T immune compartments denoted with cluster colour code and their key phenotypes. **d** Graphs showing percentages of clusters (C) 0, 4, 7 and 19 with significant enrichment in T versus P and N at three stages of HCC. Clusters phenotypes are provided on top of each graph. Boxplots show median and the whiskers represent minimum and maximum values with the box edges showing the first and third quartiles. One-way ANOVA test with two-sided $p$-values by unpaired Mann–Whitney $U$ (MWU) tests for two-group comparisons. **a–d** TNM stage I, II and III (S1, S2 and S3). $n_{PS1} = 13$, $n_{PS2} = 12$, $n_{PS3} = 9$, $n_{NS1} = 12$, $n_{NS2} = 12$, $n_{NS3} = 9$, $n_{TS1} = 13$, $n_{TS2} = 12$, $n_{TS3} = 9$. Source data are provided as a Source Data file.

S2 tumours with reduced *GZMB* (granzyme B) and enriched *PDCD1* (PD-1), *FOXP3* and *CTLA4* compared to N or S1 and S3 tumours (Fig. 3a). A consistent trend was also observed when analysed on CD4$^+$ T and CD8$^+$ T cells separately, where Treg markers and exhaustion markers were found to be enriched in S2 tumours respectively; while the proinflammatory gene *GZMB* was depleted in S2 tumours (Supplementary Fig. 5a).

From the NK cell subsets, proinflammatory cytokines such as *IFNG* and *TNF*; chemokine, *XCL1*, which could promote cDC1 recruitment critical for antitumor immunity;[21] as well as *TNFSF14*, which is important for antitumour function of NK cells via DC maturation;[22] were all depleted from S2 tumours but enriched in S1 or S3 tumours (Fig. 3b). In addition, for mononuclear phagocytes (MNPs) (Fig. 3c), we found several genes enriched in S2, such as *SPP1*, which is highly expressed in tumour-associated macrophages (TAMs) that promotes M2 polarization facilitating immune escape[23] and *TREM2*, which marks the highly immunosuppressive TAMs[24]. In contrast, *CD38*, whose expression on myeloid cells predicts favourable prognosis hence its anti-tumoral activity in HCC[25] and proinflammatory cytokine *TNF*, were both reduced in S2 and S3 tumours (Fig. 3c).

Taken together, these data further corroborate a peak of phenotypic immune evasion in S2 HCC tumours.

**Tumour co-evolution alongside immune evasion.** To better understand the onset of immunosuppression in the S2 TME, we performed transcriptomic analysis on parts of the same N and T tissues samples used for CyTOF. Principal component analysis (PCA) revealed that the greatest differences in gene expression were observed between N and T from any stage, while S2 tumours showed the greatest differences from N, S1 or S3 tumours (Fig. 4a), consistent with the most significant immune landscape changes at S2 tumours as shown above.

Next, we examined the differentially expressed genes (DEGs) in samples of the three tumour stages (Supplementary Fig. 6a). The highest number of upregulated immune-related pathways was found in S1 compared to S2 or S3 tumours (Fig. 4b, c). For instance, S1 tumours were enriched with genes involved in chemotaxis, innate immune response and cellular response to TNF, IL-1 and IFNγ; while S2 tumours were enriched with genes of the Wnt signalling pathway, which was shown to be involved in immune exclusion[26] (Fig. 4b and Supplementary Table 6). Conversely, more immune-related pathways were downregulated in S3 tumours compared to S2 or S1 tumours (Fig. 4b, c); particularly those involved in antigen presentation, T cell receptor signalling and chemotaxis (Fig. 4b, c and Supplementary Table 7). We further examined the expression of specific genes involved in these key pathways and found that antigen presentation-related genes such as the *HLA*s genes and immune response-related genes including *CD40*, *TLR7*, *GATA3* and *IL6* showed progressively reduced expression in advanced compared to early stage tumours (Fig. 4d). Interestingly, we also observed a parabolic expression in genes involved in immune exhaustion (e.g. *TCF7*, *LAG3*, *CTLA4* and *PDCD1*) (Fig. 4e), which is consistent with the

peak of immune landscape remodelling at S2 tumours as shown by our data above. Of note, *TCF7* despite being a key factor in establishing exhausted T cells[27], could also present a stem-like memory T cell phenotype associated with response to immunotherapy[28,29]. We examined TCGA HCC cohort and concluded similar downtrends of these key genes involved in the immune processes along tumour progression (Supplementary Fig. 6b).

We hypothesized that the potential partial immune recovery in S3 tumours (Fig. 2d, e) could be induced by the emergence of neoantigens detected by the immune system[11]. We quantified neoantigens in tumour samples from each stage using whole-genome sequencing (Methods) and found significantly more heterogenous (occurring in at least one but not all tumour sectors), but not ubiquitous (occurring in all tumour sectors) neoantigens in S3 than S1 tumours (Fig. 4f). This late increase in heterogenous neoantigens, a hallmark of intratumoural heterogeneity, might induce a relative increase in activated immune cell subsets observed in S3 HCC. To understand the functional implications of this partial recovery in S3 HCC, we compared the ability of these immune cells to respond to stimulation in vitro following PMA/Ionomycin stimulation. Significantly lower percentages of CD3 + T cells expressing the inflammatory cytokines IFNγ and TNFα were detected in S2 and S3 tumours than S1 tumours (Fig. 4g and Supplementary Fig. 4e). This indicates that despite the apparent partial-recovery of immunoactive subsets and the increase in heterogenous neoantigen load in late HCC, the overall immunosuppressive TME in S2 tumours likely would render this recovery functionally ineffective and unable to prevent worsening of the disease. Taken together, we showed that the transcriptomic landscape of tumour co-evolved with a progressively immunosuppressive TME.

**Immune exclusion with HCC progression.** Another possible pathway for maintaining the suppressive TME despite the apparent recovery of active immune subsets seen in advanced tumours is immune exclusion, where the TME turns immunologically "cold"[30]. This is first implicated by transcriptomic upregulation of Wnt signalling pathway in S2 tumours (Fig. 4b and Supplementary Table 6) or activation of the CTNNB1 pathway that has been linked with immune exclusion in previous studies[26,31,32]. We next measured the tumour expression of genes that were over- or under-expressed in the previously reported CTNNB1-transcriptomic signature from HCC[33] (Supplementary Fig. 7a). Indeed, we found significant regulation in genes involved in this CTNNB-1 signature, particularly in S2 tumours (Fig. 5a). Similar increase was also observed in some of the key Wnt pathway genes such as *CTNNB1*, *AXIN1* and *AXIN2* (Supplementary Fig. 7b) suggesting a peak of immune exclusion in S2 tumours.

Next, as chemotaxis is known to dictate the recruitment of various immune subsets in the TME[5,34], we further examined its relationship with the immune subsets in tumours. We selected the chemotaxis-related DEGs modified across S1–S3 tumours

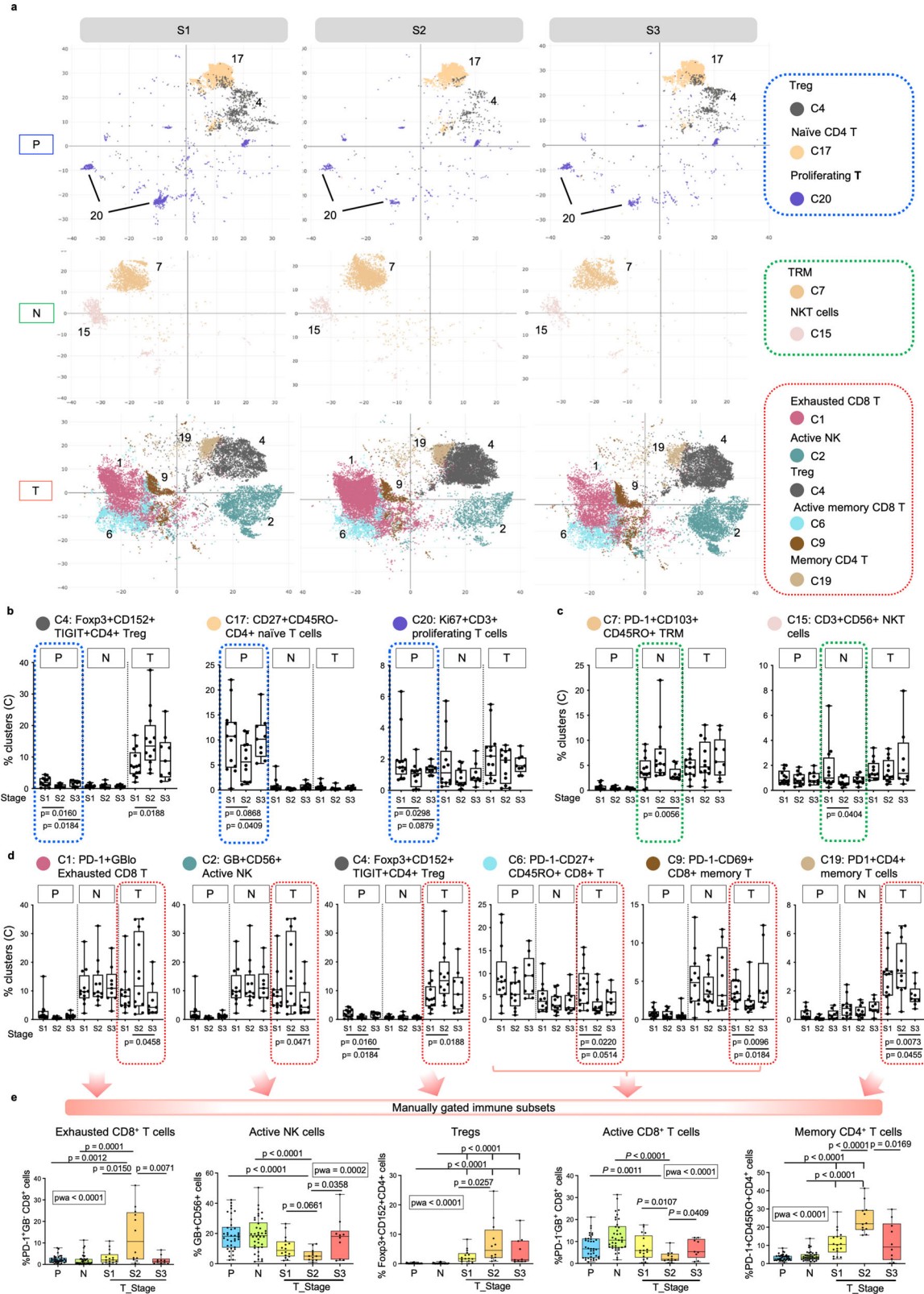

(Supplementary Table 6 and 7) and determined their correlations with the five key immune subsets altered within the tumours during HCC progression (Fig. 2e). In general, chemotactic genes that correlated positively or negatively with CD8$^+$ T cells consistently showed the opposite correlation with Treg or CD4$^+$ memory T cells, suggesting that distinct chemotactic pathways could control CD8 T cell vs immunosuppressive Treg

tumour recruitment (Fig. 5b, c). Notably, the chemokines and their corresponding chemokine receptors: *CCR1* (receptor for *CCL23* and *CCL8*), *CXCR2* (receptor for *CXCL1* and *CXCL6*) and *CXCR3* (receptor for *CXCL9*, *CXCL10* and *CXCL11*) showed a consistent trend in correlation with the key immune subsets (Fig. 5c). More importantly, we observed a progressive down-regulation of CD8$^+$ T cell chemotactic genes such as *CXCL9*,

**Fig. 2 Peak of immune evasion at stage II HCC. a** tSNE plots showing the distributions of selected immune clusters that are significantly different in abundance from PBMC (P), Non-tumour (N) or tumour (T) compartments across three TNM stages. **b** Graphs showing percentages of clusters with significant differences across TNM stages in P (blue boxes). **c** Graphs showing percentages of clusters with significant differences across TNM stages in N (green boxes). **d** Graphs showing percentages of clusters with significant differences across TNM stages in T (red boxes). **e** Graphs showing the percentages of manually gated immune populations from total live immune cells (by flowjo) in P, N and T across S1, S2 and S3 HCC. The bar and arrows indicate the major immune cell lineages shown below. **a–d** TNM stage I, II and III (S1, S2 and S3). $n_{PS1} = 13$, $n_{PS2} = 12$, $n_{PS3} = 9$, $n_{NS1} = 12$, $n_{NS2} = 12$, $n_{NS3} = 9$, $n_{TS1} = 13$, $n_{TS2} = 12$, $n_{TS3} = 9$. **b–e** Phenotypes are indicated on top. **b–e** Boxplots show median and the whiskers represent minimum and maximum values with the box edges showing the first and third quartiles. One-way ANOVA test with two-sided p-values by unpaired two-sided Mann–Whitney U (MWU) tests for two-group comparisons. **e** $n_{PS1-S3} = 38$, $n_{NS1-S3} = 38$, $n_{TS1} = 16$, $n_{TS2} = 12$, $n_{TS3} = 10$. Source data are provided as a Source Data file.

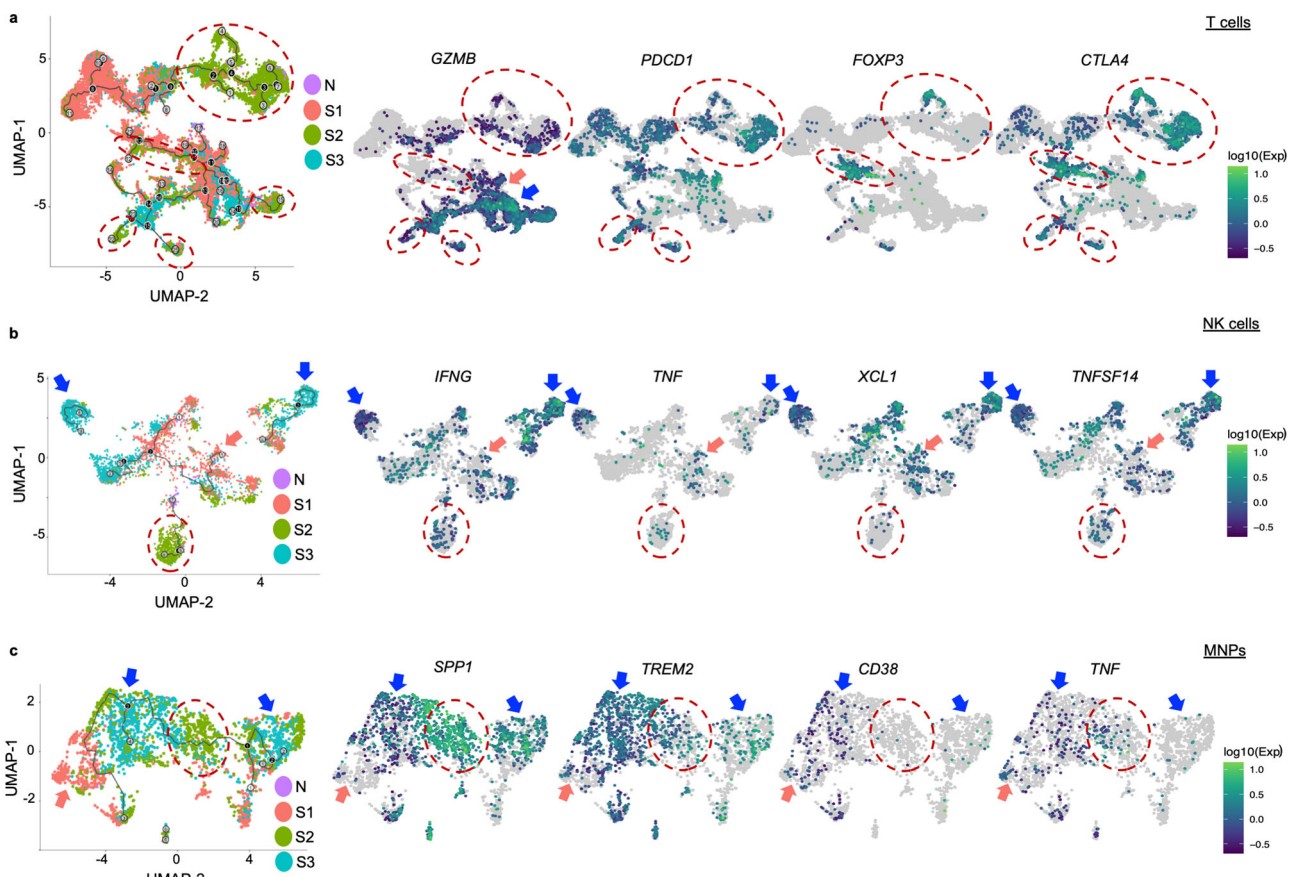

**Fig. 3 Single-cell pseudotime trajectory analysis of immune cells along HCC progression. a** Pseudotime trajectory ordering of tumour-infiltrating T cells along tumour progression. **b** Pseudotime trajectory ordering of tumour-infiltrating NK cells along tumour progression. **c** Pseudotime trajectory ordering of tumour-infiltrating mononuclear phagocytes (MNPs) along tumour progression. **a–c** Expression level of selected genes are shown. Trajectory is arranged according to the starting point of S1 tumours (n = 7), versus adjacent normal liver tissues (N) (n = 14), S2 tumours (n = 4) and S3 tumours (n = 3). S2 enriched areas are marked by red dotted line circles, and S1 or S3 enriched areas are marked by pink or blue arrows, respectively. .

*CXCL10*, *CXCL11* and *XCR1* (Fig. 5d), further indicating a progressively "cold" TME.

To validate the above observations with an actual immune exclusion, we performed immunohistochemistry (IHC) on FFPE tissues from an independent HCC cohort (n = 102 patient samples) to detect tumour infiltration of CD8[+] or exhausted PD-1[+]CD8[+] T cells. Indeed, we observed a progressive reduction in the density of total CD8[+] T cells from S1 to S3 HCC tumours (Fig. 5e, f), consistent with progressive immune exclusion. Concurrently, both the density and proportion of PD-1[+] exhausted CD8[+] T cells were higher at S2 than S1 or N (Fig. 5g), consistent with the peak of immune exhaustion shown above (Fig. 2d, e). In summary, these data show that immune exhaustion and exclusion were established and peaked at S2 and persisted into S3 tumours promoting tumour progression.

**Immune evasion trajectory validated in murine HCC model.** The human HCC data above demonstrated a dynamic pattern of co-evolution between tumour and immune compartments during HCC progression. To further validate the longitudinal process linking these stages, we employed the diethylnitrosamine (DEN)-induced HCC model, which as similar to the human condition, develops HCC from a background of chronic liver inflammation[35,36]. In these mice, early tumours at 6 months (mos) post DEN-treatment are generally small and monomorphic, while larger tumours emerge from eight to 12 mos post-injection[37] (Fig. 6a, b), representing different stages of HCC development. When we examined the TILs from early (6 mos), intermediate (8 mos) and late (12 mos) stage post-DEN tumours from these mice by flow cytometry, we found significantly higher proportions of Foxp3[+]CD25[+]CD4[+] Treg and

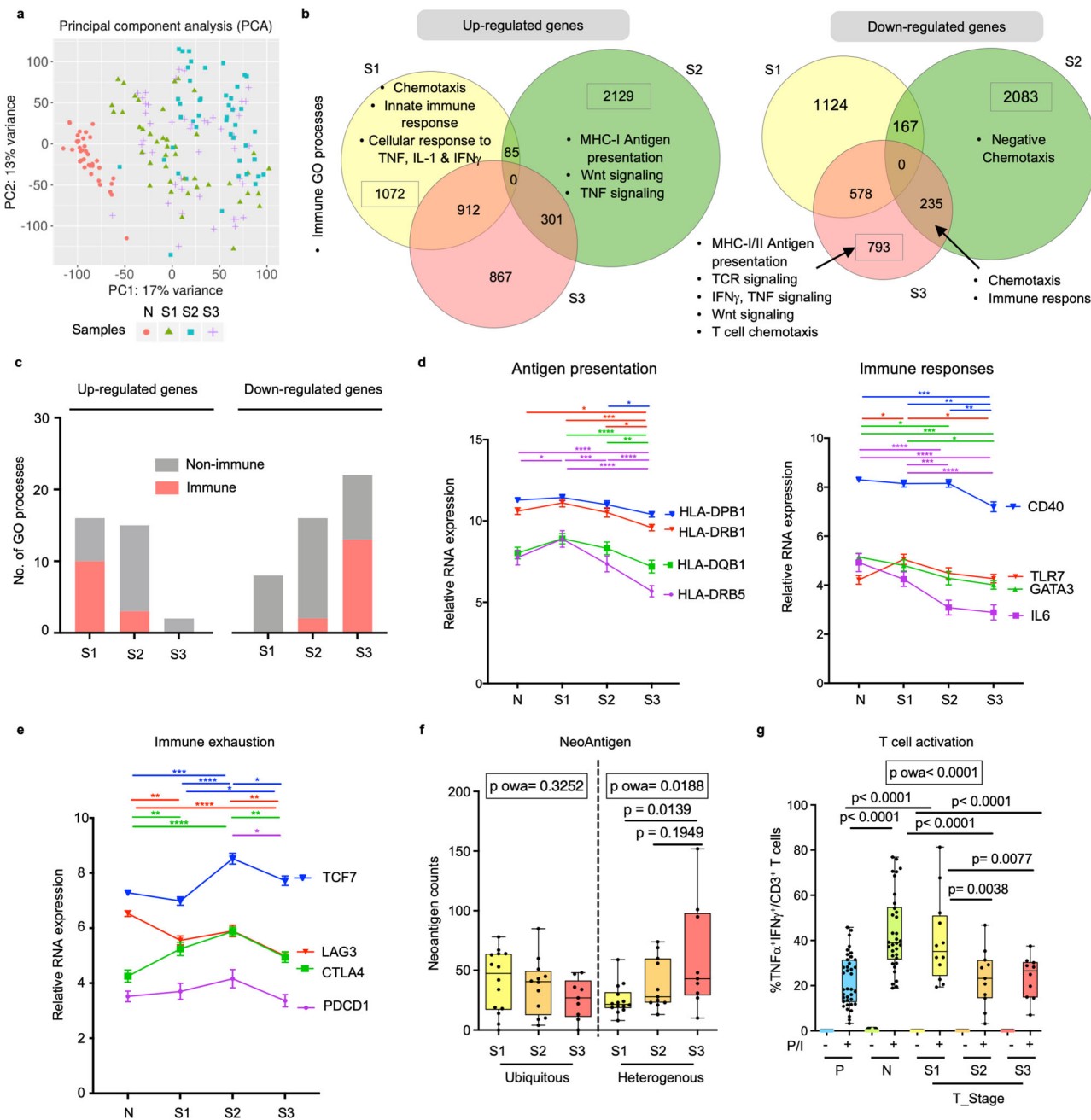

**Fig. 4 Progressive tumour co-evolution along immune evasion at different HCC stages. a** Principal component analysis (PCA) of RNA sequencing profiles reveals sample distribution of HCC tumours from TNM stage I, II and III (S1, S2 and S3) and non-tumour tissues (N). **b** List of enriched immune-related Gene Ontology (GO) processes among tumours from three TMN stages. **c** Graphs compare the number of immune or non-immune related GO processes indicated by the upregulated or downregulated genes across different tumour stages (S1, S2 or S3). **d** Expression levels of genes involved in antigen-presentation (left) and immune responses (right) during tumour progression. **e** Expression levels of genes involved in immune exhaustion during tumour progression. **d**, **e** Two-way Anova test followed by respective unpaired Tukey's pairwise comparison test colour coded by each gene; *, **, ***, **** denotes two-sided $p$-values <0.05, <0.01, <0.001 and <0.0001 respectively. Graphs show mean ± standard error of the mean. $n_N = 37$, $n_{S1} = 43$, $n_{S2} = 43$ and $n_{S3} = 45$. **f** Changes of ubiquitous and heterogenous neoantigen counts across three tumour stages. $n_{TS1} = 14$, $n_{TS2} = 12$, $n_{TS3} = 9$. **g** Percentages of inflammatory TNFα+IFNγ+ active CD3+ T cells from peripheral blood (P) ($n = 38$), N ($n = 34$) and S1 ($n = 12$), S2 ($n = 11$) and S3 ($n = 10$) tumours (T_Stage), upon stimulation with PMA/Ionomycin (P/I) for 5 h. **f**, **g** Boxplots show median and the whiskers represent minimum and maximum values with the box edges showing the first and third quartiles. One-way ANOVA test (p owa) with two-sided $p$-values calculated by Tukey post-hoc multiple pairwise comparisons. Source data are provided as a Source Data file.

PD-1+ exhausted CD8+ T cells, alongside significantly lower proportions of CD69+ active CD8+ T cells and NK 1.1+ NK cells at intermediate stage tumours at 8 mos post DEN induction (Fig. 6c, d and Supplementary Fig. 8a). This supports our human

data, where tumours evaded the immune response progressively and peaked at the intermediate tumour stage.

We also detected PD-1hi, PD-1lo and PD-1− populations among CD8+ T cells with a significant accumulation of the

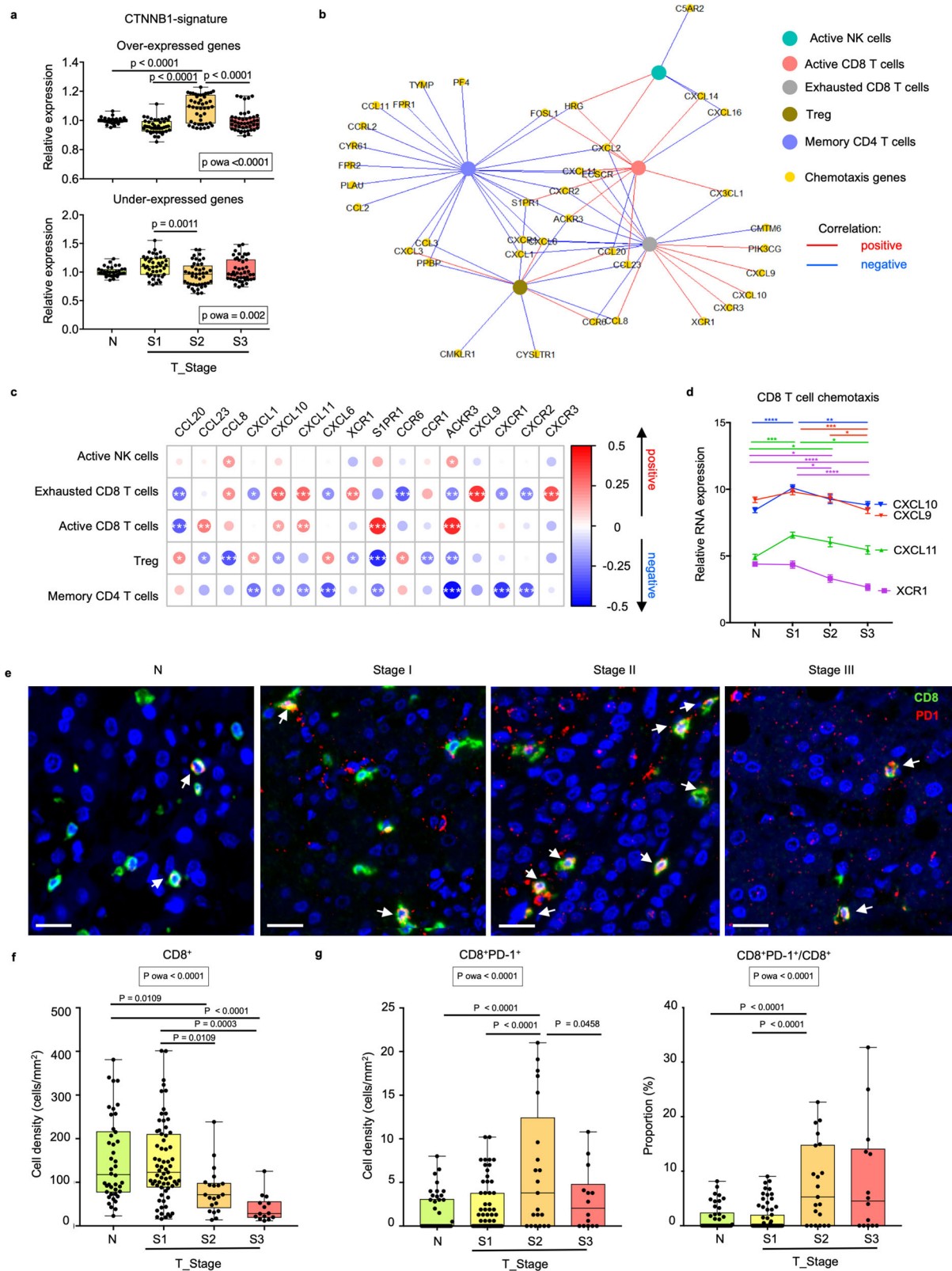

PD-1$^{hi}$ and depletion of PD-1$^{lo}$ population within CD8$^{+}$ T cells from intermediate 8 mos post-DEN HCC tumours compared to 6 or 12 mos post-DEN tumours (Fig. 6e, f). This again supports the immune evasion peaking at intermediate HCC. As PD-1$^{hi}$ CD8$^{+}$ T cells are linked with immune exhaustion and are the prime target of immune-checkpoint blockade treatment for HCC[38],

these data support the potential benefit of immunotherapy in intermediate HCC.

For further validation of immune exclusion on these DEN-induced tumours, we performed IHC on tumour tissues from 4, 8 and 12 mos post-DEN induction and indeed found consistent trend of progressive immune exclusion with decreasing CD8$^{+}$ T

**Fig. 5 Immune exclusion from stage II HCC. a** Expression levels of over-expressed (top) or under-expressed (bottom) genes in beta-catenin (CTNNB1) signature in adjacent non-tumour liver tissues (N) ($n_N = 37$) and tumours from different stages ($n_{S1} = 43$, $n_{S2} = 45$ and $n_{S3} = 45$). **b** Chemokine genes (yellow circles) versus immune subsets (multiple colours circles) network showing positive (red lines) and negative (blue lines) correlations (Spearman's correlation test, rho $\geq 0.2$, $P < 0.05$). **c** Heatmap shows positive (red circles) and negative (blue circled) correlations between 16 selected chemotactic genes (columns) and five immune subsets (rows). Circle sizes represent Spearman's rho; *, **, ***, **** denotes p-values < 0.05, <0.01, <0.001 and <0.0001 respectively. **d** Expression levels of genes involved in CD8 T cell related chemotaxis. Two-way ANOVA test followed by respective unpaired Tukey's pairwise comparison test colour coded by each gene; *, **, ***, **** denotes two-sided p-values < 0.05, <0.01, <0.001 and <0.0001, respectively. Graph shows mean ± standard error of the mean. $n_N = 37$, $n_{S1} = 43$, $n_{S2} = 43$ and $n_{S3} = 45$. **e** Representative immunohistochemistry (IHC) images of CD8 (green) and PD-1 (red), respectively in N and S1-S3 HCC tumours from an independent validation cohort. CD8$^+$PD-1$^+$ populations are indicated by white arrows. Bar = 20 um. **f** Boxplots compare densities of CD8$^+$ T cells and **g** densities of CD8$^+$PD-1$^+$ T cells (left) as well as proportions of PD-1$^+$CD8$^+$/ CD8$^+$ T cells (right) by IHC in N and tumours from S1-S3 stages. **f**, **g** $n_N = 42$, $n_{S1} = 67$, $n_{S2} = 21$ and $n_{S3} = 14$. **a**, **f** and **g** Boxplots show median and the whiskers represent minimum and maximum values with the box edges showing the first and third quartiles. One-way ANOVA test (p owa) with two-sided p-values calculated by Tukey post-hoc multiple pairwise comparisons. Source data are provided as a Source Data file.

cell density from early to late stages tumours (Fig. 6g, h). Interestingly, the peak of density in CD8$^+$PD-1$^+$ populations occur consistently at intermediate 8 mos post-DEN tumours (Fig. 6h).

Taken together, tumours from this HCC murine model recapitulate our earlier findings in human HCC of an ongoing immune evasion peaking at intermediate tumour stage.

## Discussion

In the current study, we have delineated the immune modifications across different tissue compartments at various stages of treatment naïve HCC, thereby uncovering strategic points for therapeutic intervention and advancing our understanding of the complex interplay between tumour cells and the immune system. In contrast to the current dogma that immune suppression/evasion occurs either early during carcinogenesis or late prior to metastasis, we found that in HCC, it is a progressive and continual process that peaks at intermediate stage II tumours. This peak is characterised by relatively higher frequencies of exhausted and suppressive immune subsets and reduced frequencies of activated immune subsets; concurrently, the immune landscape turns "cold", with reduced numbers of infiltrating CD8$^+$ T cells, rendering tumours refractory to immune-mediated suppression. This immune evasion could be due to the modifications in the tumour transcriptomic landscape, which concurrently mirrored the immune microenvironment, with lower expression of genes involved in several immune-related pathways as well as upregulation of CTNNB1-related genes associated to immune exclusion, both occurred at S2 HCC tumours.

Interestingly, we found potential evidence of partial immune recovery in stage III tumours corresponding with an increase in neoantigens, which have previously been shown to induce immune responses in the TME[39]. However, given the fact that most advanced HCC patients may not undergo resection as first-line therapy, these interesting findings will require careful interpretation against this inevitable confounding factor. Furthermore, the small recovery of immune cell subsets in stage III HCC tumours was coupled with overall lower immune activation status as well as immune exclusion: thus, the resurgence of immuno-surveillance at this stage may not significantly impact disease progression. Nevertheless, these findings raise the possibility that immunotherapy, if timed carefully to exploit neoantigen exposure/immune resurgence in stage III tumours, could be a potentially powerful intervention in late-stage HCC, supported also by the efficacy of immune checkpoint blockade in advanced HCC[40,41]. The efficacy of immune checkpoint inhibitors, as well as Wnt inhibitors, remain an interesting area for further research.

In colorectal cancer, a similar inverse relationship between tumour-infiltrating CD8$^+$ T cells and advanced tumours was observed[42]. In contrast, an in silico transcriptomic study in HCC failed to detect significant changes in immune composition across the various tumour stages[43]. The discrepancy in the later study might originate from technical differences between bulk transcriptomic deconvolution used in this previous study versus the comprehensive single-cell immunoprofiling performed in the current study. On the other hand, the concept of tumour evolution and the important role of immune contexture in immuno-surveillance or immunoediting have been discussed elsewhere[44]. Immune evolution in HCC was proposed in our recent study to be manifested as immune intratumoural heterogeneity (immune-ITH)[14]. In addition, single-cell transcriptomic analysis of TILs in HCC also revealed immune evasion with T cell exhaustion[45], myeloid/lymphoid cross-interaction[46] and onco-foetal reprogramming of endothelial cells and macrophages[19]. In this study, we revealed an unprecedented progressive immune evolution alongside tumour progression peaking at the intermediate stage of HCC.

Longitudinal studies of tumour progression and its relationship to the immune landscape in human colorectal and lung cancer do suggest that evolution of these tumours is immune-related or immune-driven[10,12], at least at the very early or very late disease stages captured in these studies. Similar to other human solid cancer studies, our current study has been limited to inferring the parallel cancer and immune evolution using specimens taken from different patients at various stages of tumour[11,47,48]. To demonstrate the trajectory of tumour landscapes evolution, both scRNA pseudotime trajectory prediction on tumour-infiltrating immune cells as well as early to late stages DEN-induced HCC model validated our discovery on a peak of phenotypic immune evasion at the intermediate tumour stage and a progressively immune excluded TME during tumour progression.

To conclude, our in-depth analysis of intratumoural immune dynamics during HCC progression has profound implications for our understanding of HCC evolution and will help guide the development of potential immunotherapeutic treatments.

## Methods

**Patient samples**. The study was approved by the Central Institution Review Board (CIRB) of SingHealth, of which all National Cancer Centre Singapore, Singapore General Hospital and National University Hospital were constituent members (CIRB Ref: 2016/2626 and 2018/2112). Tumour (T), adjacent non-tumour liver tissue (N) and peripheral blood (P) were collected from 38 HCC patients, each provided written informed consent, with demographics and clinical characteristics as described in (Supplementary Table 1). For each patient, a single tumour slice was further dissected into 2 to 5 sectors (depending on the size of the tumour) with at least 1 cm separation between them. Adjacent non-tumour liver tissue from at least 2 cm away from the tumour was also harvested. Each tissue sector was divided for analysis by CyTOF, RNA and whole genome sequencing. Peripheral blood mononuclear cells (PBMC) ($n = 38$) were isolated by Ficoll-Paque Plus (GE Healthcare) density gradient centrifugation. Tumour-infiltrating leucocytes (TIL) ($n = 135$) and non-tumour tissue-infiltrating leucocytes (NIL) ($n = 38$) were isolated from each tissue separately using enzymatic digestion with 500 μg/mL collagenase IV (Thermo Fisher Scientific; Cat#: 17104019) and 50 μg/mL DNase I

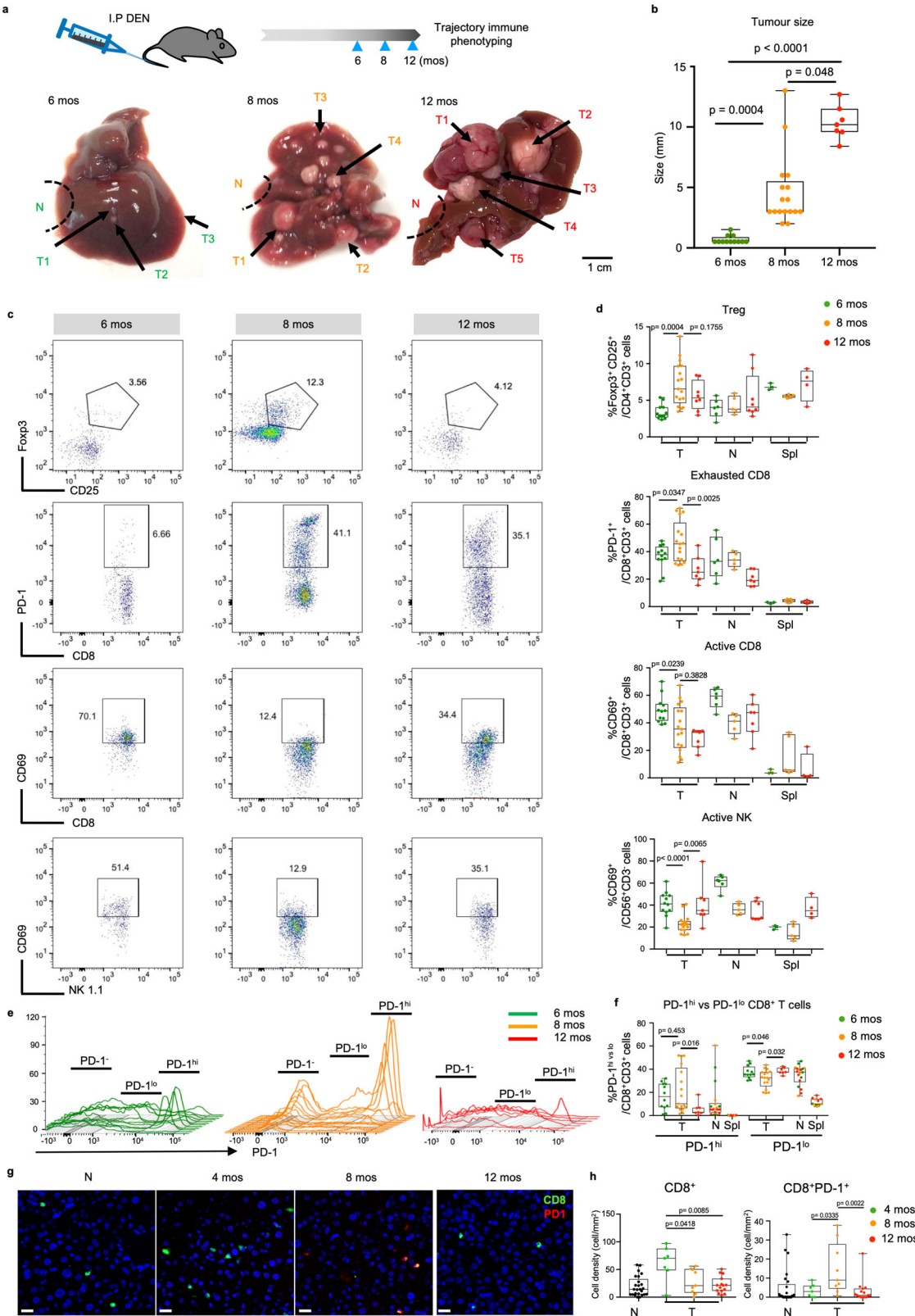

(Roche, Indianapolis, IN; Cat#: 4716728001) for 30 min in 37 °C. Cells were stored in liquid nitrogen with 10% DMSO in foetal bovine serum (FBS) until further analysis by CyTOF.

**Cytometry by time of flight (CyTOF).** TILs, NILs and PBMCs were thawed and rested for at least 30 min in RPMI medium with 10% FBS and 1% penicillin/streptomycin. Only samples with >100,000 live cells after thawing will be subjected

to downstream CyTOF analysis (note: only two tumour sectors with insufficient cell numbers were excluded from the entire cohort). Data was collected from at least two sectors of each tumour in this study. The cells were unstimulated or stimulated with 150 ng/mL PMA (phorbol myristate acetate; Sigma) and 100 ng/mL ionomycin (Sigma) and incubated for a total of 5 h in 37 °C and 5% $CO_2$ and 3 μg/mL Brefeldin A (eBiosience) and 2 μM monesin (BioLegend) was added at the last 3.5 h of the incubation. The cells were then washed and stained with cisplatin-Live/Dead stain (Fludigm) before labelling with a combination of barcoding anti-

**Fig. 6 Validation of immune evasion during tumour progression in murine HCC model. a** Representative images of murine livers bearing multiple tumours (T) from 6, 8 or 12 months (mos) after DEN-induction. Bar = 1 cm. **b** Tumours from 6, 8 or 12 mos after DEN-induction showing varying tumour size. **c** Representative dot plots of Foxp3+CD25+ Treg, PD-1+ exhausted CD8 T cells, CD69+PD-1− active CD8 T cells and CD69+NK1.1+ active NK cells in 6, 8 or 12 mos post-DEN tumours. **d** Comparison of frequencies (%) of Treg, exhausted CD8 T cells, active CD8 T cells and active NK cells in 6, 8 or 12 mos T, adjacent non-tumour liver tissues (N) and spleen (spl). **e** Histograms showing relative expression of PD-1 further distributed to PD-1hi, PD-1lo and PD-1− CD8+ T cell populations from 6 mos (green lines), 8 mos (orange lines) and 12 mos (red lines) post-DEN induced murine HCC tumours. **f** Graph showing percentages of PD-1hi and PD-1lo CD8+ T cell populations across 6, 8 or 12 mos post-DEN T, N and spl. **g** Representative immunohistochemistry (IHC) images of CD8 (green) and PD-1 (red), respectively in N and murine HCC tumours from 4, 8 or 12 mos post-DEN induction. Bar = 20 um. **h** Boxplots compare densities of CD8+ T cells and CD8+PD-1+ T cells by IHC in N ($n = 25$) or T from 4 ($n = 7$), 8 ($n = 9$) or 12 ($n = 15$) mos post-DEN induction. **b, d, f** and **h** Boxplots show median and the whiskers represent minimum and maximum values with the box edges showing the first and third quartiles. Non-parametric one-way ANOVA Kruskal–Wallis test with two-sided p-values calculated by Dunn's post-test between groups. **b, d** and **f** graphs show data from $n_{T6mos} = 12$, $n_{T8mos} = 17$, $n_{T12mos} = 7$, $n_{N6mos} = 6$, $n_{N8mos2} = 5$, $n_{N12mos} = 7$, $n_{spl6mos} = 3$, $n_{spl8mos} = 5$, $n_{spl12mos} = 4$ post-DEN murine samples. Source data are provided as a Source Data file.

CD45 antibody conjugated to three different metals, for simultaneous analysis of multiple samples[49]. Cells were then combined and labelled with metal-conjugated antibodies targeting surface markers before fixing with 1.6% paraformaldehyde, permeabilizing with 100% methanol and staining with metal-conjugated antibodies targeting intracellular molecules. All 33 antibodies and barcoding anti-CD45 antibodies were either conjugated in-house according to the manufacturer's instructions (Fluidigm) or purchased pre-conjugated directly from Fluidigm (Supplementary Table 2). Finally, an iridium-containing DNA intercalator was added to identify single cells and the cells were washed and diluted with EQ Four Element Calibration Beads (Fluidigm) for signal normalization according to manufacturer's instructions[50], before data were acquired using Helios equipped with the CyTOF® 6.7 system control software (Fluidigm).

The generated files were analysed by FlowJo (v.10.2; FlowJo): live single-cells (cisplatin-negative and DNA-intercalator-positive) were debarcoded to each sample file based on their unique CD45 barcodes. Each file was then down-sampled to 10,000 cells and further analysed using our in-house Extended Polydimensional Immunome Characterization (EPIC) analysis pipeline with data visualization using the browser-based R shiny app 'SciAtlasMiner'[16]. We also performed batch normalization, data visualization and statistical tests[16]. Clustering was performed using the Phenograph algorithm (v1.5.2)[15] or FlowSOM algorithm[17] and dimension reduction using tSNE. To compare the frequencies of cell populations between tissue types and clinical HCC stages, we applied one way ANOVA test with post-pairwise Mann–Whitney U-test. Of note, we averaged cluster frequencies from multiple tumour sectors to represent each patient for all statistical tests. Validation using manual gating was performed with FlowJo (v.10.2). Immune diversity was quantified by calculating the multivariate beta-dispersion of the Bray–Curtis distances between immune clusters for each tissue type using the 'vegan' R package v2.5. The beta-dispersion were compared among tissues and p-values were determined using Tukey's HSD (honestly significant difference) test.

**Single-cell RNA sequencing.** Single cells were isolated from tumour and adjacent non-tumour tissues from 14 patients with S1 ($n = 7$), S2 ($n = 4$) and S3 ($n = 3$) HCC, followed the same enzymatic digestion as described previously[19]. Briefly, dead cells were removed using dead cell removal kit (Miltenyi, Cat# 130-090-101). CD45+ cells were enriched using CD45 MicroBeads (Miltenyi, Cat#: 130-045-801) before CD45+ and CD45− cells were processed using the Chromium Single Cell 30 (v2 Chemistry) platform (10x Genomics, Pleasanton, CA) as previously described[19]. All data were then aggregated using cellranger aggr by normalizing all runs to the same sequencing depth. Downstream analysis was performed using Scanpy (version 1.4; all genes expressed by a minimum of 30 cells were considered and cells with <200 genes and >5% mitochondrial content were excluded from analysis). For clustering, best matched k-Nearest Neighbour is automatically weighted by the algorithm to compute the best UMAP topology (scanpy.api.tl.umap, minimum distance between 0.3 and 0.5). The Louvain method (scanpy.api.tl.louvain) is then used to detect a community of similar cells resolution parameter 0.6 to 1 as illustrated previously[19]. Pseudotime trajectory analysis of tumour-infiltrating immune cells were performed using the Monocle R package (version 3.0)[20], on T cells (further sub-divided to CD3+CD8- and CD3+CD8+ T cells), NK cells and MNPs clusters. A Cell Data Set object was created according to Monocle's pipeline. Log-normalisation and scaling was done using Monocle's pre-processing function. Principle Component Analysis (PCA), dimension reduction and clustering were performed using the default parameters. Pseudotime trajectories were plotted for each partition using Monocle's reversed graph embedding algorithm. For pseudotime ordering, cells from the Stage 1 (S1) tumour samples were taken as the starting point.

**Immunohistochemistry (IHC) tissue staining.** Formalin-fixed paraffin-embedded (FFPE) sections were from an independent cohort of 102 HCC patients who underwent curative resection from 1991 to 2009, and were obtained from the Department of Anatomical Pathology, Division of Pathology, Singapore General Hospital. Two cores of tumour and non-tumour FFPE tissues from each patient

were stained with Opal™ Multiplex immunohistochemistry Detection Kit and images were acquired using Vectra 3.0 pathology imaging system microscope (Perkin Elmer) according to manufacturer's instructions. We used antibodies: anti-CD8 (DAKO, Clone:C8/144B, 1:200) and anti-PD-1 (Abcam, Clone: NAT105, 1:100) and detection dye: Opal690 dye (CD8) and Opal650 dye (PD-1) with DAPI as nuclear counterstain. Quantification of stained cells was performed on the whole 1 mm core (Area = 0.785 mm²) and the average values of two cores from each patient sample was calculated and shown as the number of cells/mm². Separately, IHC staining was done on FFPE tissues from DEN-induced mice (See description below). Adjacent non-tumour (N) and tumour samples from 4 mos ($n = 7$), 8 mos ($n = 9$) and 10 mos ($n = 9$) post-DEN mice were stained with Anti-CD8 (Abcam, Catalogue #ab203035; polyclonal; 1:500), anti-PD-1 (Abcam; clone EPR20665; 1:1000) and detection dye: Opal650 dye (CD8) and Opal620 dye (PD-1) with DAPI as nuclear counterstain. Quantification was performed and analyzed from at least 2–6 (limited by tumour size) random 0.6 × 0.5 mm² regions.

**Bulk RNA sequencing.** Total RNAs from multiple tumour sectors and each non-tumour sector were isolated using Picopure RNA-Isolation kit (Arcturus, Ambion), and cDNA was generated using the SMART-Seq® v4 UltraTM Low Input RNA Kit for Sequencing (Clontech, USA). Illumina indexed libraries were created using the Nextera XT DNA Library Prep Kit (Illumina, USA) and multi-plexed for 2×101 bp-sequencing. RNA sequencing was performed on a HiSeq High output platform at the Genome Institute of Singapore (GIS).

The raw reads were aligned via STAR[51] to the Human Reference Genome hg19, and the gene-level expected counts were calculated using RSEM[52]. Only protein-coding genes with at least one count per million reads in 5% or more of the samples were retained, and the data was normalized using DEseq2[53]. Differential expression analysis was performed using R package Limma[54] with the false discovery rate (FDR) adjusted for multiple testing using Benjamini–Hochberg procedure[55]. Differentially expressed genes (DEGs) were identified at an FDR < 0.05 with pairwise comparison. Pathway enrichment analyses were performed using integrated Differential Expression and Pathway analysis (iDEP)[56] v0.91 and DAVID pathway analysis v.6.8, both with adjusted p-value < 0.01. For validation of selected genes of interest, the raw counts for the Liver HCC, The Cancer Genome Atlas (TCGA) dataset were downloaded from FireBrowse[57]. Only protein-coding genes with raw counts > 1 (TCGA dataset) in ≥5 samples were retained and data were normalized using DEseq2 ($n = 297$: S1, $n = 151$; S2, $n = 75$ and S3 $n = 71$).

Genes over- or under-expressed in CTNNB1-mutation signature were previously reported in HCC[33], and the relative expression levels of these genes which were calculated for each tumour according to the median expression of each gene. The expression levels of genes of interest were compared among three tumour stages using one-way ANOVA test with post-hoc Tukey's pairwise test. Correlation between genes and immune subsets were computed with Spearman's rho and p-values and visualized using R package "corrplot": Visualization of a Correlation Matrix (Version 0.84) and "network": Butts C (2020).

**Whole genome sequencing and neoantigen prediction.** DNA from tumour and adjacent non-tumour tissues was extracted using Qiagen AllPrep kit and sonicated into shorter fragments using the Covaris system. Quality check was performed using Agilent 2100 Bioanalyser. DNA fragments were end-repaired, ligated, amplified, and sequenced using the Illumina sequencing platform at GIS. Raw reads were mapped to the Human Reference Genome hg19 using the Burrows–Wheeler Aligner[58]. After removal of duplicated reads by PICARD (http://broadinstitute.github.io/picard/), base quality recalibration and realignment were performed using the Genome Analysis Tool Kit[59]. Comparison of tumour against non-tumour tissue was performed using Mutect (version 1.1.7)[60] to call for somatic variants.

Neoantigen prediction was achieved using personalized Variant Antigens by Cancer Sequencing(pVacSeq)[61], with variant calling information obtained from MuTect[60]. The variant calls were annotated using VEP: 8- to 11-mer epitopes with

<500 nM predicted binding affinity to MHC-class 1. The ubiquitous (expressed by all tumour sectors) and heterogenous (expressed by at least one but not all sectors) neoantigens as defined previously[62] were computed.

**DEN-induced mouse model of HCC.** The animal procedure was approved by and conducted in accordance with the regulations and guidelines of the Institutional Animal Care and Use Committee (IACUC) of the National University of Singapore and SingHealth Institute (IACUC reference number: R019-0666 and 2016/SHS/1228). HCC was induced by intraperitoneal injection of a single dose of diethylnitrosamine (DEN) (Sigma-Aldrich) at 25 mg per Kg body weight to 15 days old wild-type male C57BL/6 J mice, consistent with previously published method[37]. Mice were weaned at 21 days of age and maintained on the Chow diet and 12 h light/12 h dark cycles at 23 °C co-housed in pathogen-free animal facilities. Mice were euthanized by carbon dioxide-induced asphyxia. Tumours of various sizes corresponding to different stages of development were harvested at 4 or 6 mos (early stage), 8 mos (intermediate stage) and 12 mos (late stage) post DEN induction. Immune cells isolated from tumours, adjacent non-tumour liver tissues and spleens were analysed with flow cytometry (as detailed below). Additionally, FFPE tissues from adjacent non-tumour and tumour specimens were separately analysed with IHC (as detailed above). Tumours showed varying sizes with the median diameters as 3 mm.

**Flow cytometry.** Immune cells were isolated from mouse liver tumours or healthy livers by digestion with 500μg/mL collagenase IV (Thermo Fisher Scientific) and 50 μg/mL DNase I (Roche, Indianapolis, IN) and spleen from mechanical isolation. The isolated cells were analysed using antibodies listed in Supplementary Table 8. All flow cytometry analyses were done on a BD Fortessa flow cytometer equipped with five lasers (BD Biosciences, San Jose, CA), and the data were processed using FlowJo (v.10.2).

**Statistics.** Statistical comparisons of cell frequencies between groups were performed using the parametric one-way ANOVA test (OWA) with Tukey post-hoc multiple comparison test (human data) or unpaired two-sided Mann–Whitney U (MWU) tests for pairwise comparisons and non-parametric one-way ANOVA Kruskal–Wallis test with Dunn's post-hoc multiple comparison test (murine data), both with two-sided $p$-values, in GraphPad Prism (V7.0d). The edgeR package[63] was used for differential expression analysis of the RNA-seq data. Spearman's correlation analysis was performed with the rho and $p$-values reported.

**Reporting summary.** Further information on research design is available in the Nature Research Reporting Summary linked to this article.

## Data availability

The WGS and RNA-seq data generated in this study are deposited in European Genome-phenome Archive (EGA) under the accession code: EGAS00001003814. Single-cell RNA sequencing data were deposited under the accession code GSE156625. The remaining data are available within the Article, Supplementary Information or source data provided with this paper. Source data are provided with this paper.

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

## Acknowledgements

The authors would like to thank all members of TII, all participating patients, the clinical research coordinators from NCCS, SGH, NUHS for their contributions and Dr Lucy Robinson of Insight Editing London for scientific and language editing of this manuscript. This work was supported by the National Medical Research Council (NMRC), Singapore (ref numbers: NMRC/TCR/015-NCC/2016, NMRC/CIRG/1460/2016, NMRC/CSA-SI/0013/2017, NMRC/CSA-SI/0018/2017, NMRC/OFLCG/003/2018, NMRC/STaR/020/2013, NMRC/CG/M003/2017, LCG17MAY004 and NMRC/OFIRG/0064/2017) and National Research Foundation, Singapore (ref number: NRF-NRFF2015-04).

## Author contributions

P.H.D.N. and M.W. obtained and analysed data and prepared the paper; C.T.T. performed the mouse experiments, analyzed the data and discussed the paper. C.J.L. processed the samples, performed flow cytometry and IHC experiments. H.H.L.L. C.Z.J.P. and S.M. analysed the genomic/transcriptomic data. J.J.W.S. and R.D. obtained and analyzed the scRNA seq data. J.Ya. performed the mouse experiments. S.D.S. processed the samples, performed flow cytometry and IHC experiments. W.L.T. provided genomic/transcriptomic analysis support. T.K.H.L., J.Ye., W.Q.L., Y.H.P., G. S., T.J.L. and W.K.W., prepared and provided tissue samples and discussed the data. C.Y.C., P.C.C., H.C.T., A.K., Y.Y.D. J.H.K., S.I., K.M., A.C., G.K.B. and B.K.P.G. recruited patients, provided samples and discussed the data. N.F. and V.C.Y. provided support for mouse model work. W.Z. designed and led the genomics analysis and discussed the data. S.A. designed the CyTOF pipeline and discussed the data. P.K.H.C. initiated the liver cancer programme, provided patient samples and discussed the data; V.C. designed and led the study, performed the analysis and prepared the paper.

## Competing interests

The authors declare no competing interest.
