## [Peer Review File · Nature Communications]

Trajectory of immune evasion and cancer progression in hepatocellular carcinomaEditorial Note: Parts of this Peer Review File have been redacted as indicated to remove third-party material where no permission to publish could be obtained.

REVIEWER COMMENTS

Reviewer #1 (Remarks to the Author):

In the paper by Nguyen P, et al, the authors comprehensively examined the immune subset changes in the different stages of HCC, by utilizing cutting-edge technologies, CyTOF, single cell RNA sequencing and exome sequencing. They found that the immunosuppression and exhaustion peaked in patients at stage II. The results of this research are of interesting and very important point to be addressed for future HCC therapy. However, some substantial issues are raised for the paper, especially in the study design and interpretation of the results.

Major points:

1. One of the major drawbacks of the study is the case analyses were performed only cross sectionally but not longitudinally. The authors categorized the patient group according to the TNM staging system. Although this classification is of clinical importance, patients background, such as tumor numbers, localizations, or vascular invasions, and liver function reserve, should be varied significantly even in patients in the same TNM stage. It is arguably interesting to see that the derangement of immune system peaked at the stage II but not at more advanced stage III. All the data provided by the authors, including a mouse model of chemical carcinogenesis (DEN-induced cancer), seemingly support for the results obtained from HCC patients. However, authors fail to provide some evidence, or at least some clues, to explain why such immune dysregulation peaked in the middle or intermediate stage of HCC patients. Do the authors think that CTNNB1 signature is a key? Are there any cases that the authors could examine the immune subset changes as shown here in some patients longitudinally, in the process of development from Stage I to II or III? And what are most significant factors, T, N or M, impacting on immune exhaustive features in patients?
2. The authors utilized DEN-induced carcinogenesis model as a backup for confirming the changes of immune cell alterations according to the tumor progression. Because it is well acknowledged that DEN is a chemical carcinogen and induce liver tumors, the gene signature of which is completely different from that of human HCC. In addition, how the authors define the mice at 8month should be compatible with HCC patients at stage II? The authors need to explain the rationale for using DEN-model for the comparison.

Minor

1. In mice, it is important to see the efficacy of immune checkpoint inhibitors with/without Wnt inhibitors at different time points.
2. The single cell sequence analysis in Figure2e may be useful to identify the downstream or upstream of S2 immune cell subset.
3. Same as Figure4c, are there any significance on receptors for chemotaxis?

Reviewer #2 (Remarks to the Author):

Nguyen et al characterized immune microenvironment in a series of 38 HCC with TNM stage I to III. They performed CyTOF analyses of tumor, non-tumor liver and blood samples for each patient. They identified immune depletion mainly in stage 2 patients. Immune evasion was validated in mice DEN model

Criteria of selection of the patients should be better described. In particular, in sup table 1 and in the text it is mentioned that patients with stage I, II or III were comparable. However, in sup table 1 it appears that stage II tumors were more aggressive with more MVI and proliferation. Why? Does patients with stage III are highly selected as treatable by surgery? This is a major point to correctly interpret the "immune recovery" observed in stage III patients.

In the analyses, cirrhotic patients should be compared to non-cirrhotic one. Also, in sup table 1 F4 patients should be described separately.

Since the majority of the patients are HBV infected, the precise status of HBV infection should be included in the analysis. In particular antiviral therapy and viral infection activity should be described and correlated with the immune status.

Figure 1c: stage II and stage III peripheral typing seems exactly SIMILAR

Neo-antigen load in stage II and III: are they significantly different at figure 3f?

Sup figure 5 list: over and under-expressed genes in CTNNB1 mutated tumors are unusual compared to previous publications. Please explain why?

Reviewer #3 (Remarks to the Author):

This manuscript seeks to characterize changes in immune cell populations and immune functions in hepatocellular carcinoma (HCC) across different disease stages. Using mass cytometry (CyTOF), the authors analyze immune clusters in HCC and normal liver samples, as well as peripheral blood. The authors report, stage 2 HCC exhibited lower levels of active NK cells and active memory CD8+ T-cells and higher levels of immunosuppressive CD8+ T-cells, T-reg and exhausted memory T-cells. Using IHC/IF, the authors identified the highest density of proportion of PD-1+ CD8+ T-cells in stage 2 tumors. Single-cell RNA sequencing of tumor samples with trajectory analysis was reported to show reduced CD69 expression and increased PD-1, FOXP3 and CTLA4 expression in stage 2 HCC as compared to other tumor stages. In bulk RNA sequencing and pathway analysis, stage 2 disease presented with the highest expression of genes related to immune exhaustion while the expression of genes related to immune response and antigen presentation was lower in advanced disease.

This manuscript potentially represents a relevant contribution to the literature as it may help to better understand the shifts occurring within the tumor immune microenvironment in HCC across disease stages. A “cold” TME is identified in stage 2 disease and partially validated through the different analyses performed, from mass cytometry and IHC, to bulk and single-cell RNA sequencing.

However, there are still many issues that would need to be addressed:

- Methods: more details needed. Examples include for enzymatic dissociation, how long was this performed for? Please include in methods additional details, such as catalog numbers for enzymes, timing of dissociation. What are the specific products used (catalog numbers for enzymes)?
- For patient, list each sample, and how many cells were included in the final analysis?
- Were any samples filtered out for QC reasons (i.e. low viable cell numbers)? If so, how many samples, and what were the criteria for this. This should be included in the methods.
- What is the representation of each sample in each cluster? Are certain clusters patient specific (or compartment-specific)? Please include in the additional figures to address this, such as stacked bar graphs for each cluster by tissue origin and by patient origin.
- The recovered immune population somehow appear off. First, compared to the work of Zhang et al, Cell, 2019, there appear to be many fewer myeloid cells. In reviewing the feature plots in the supplement, PD-L1 is not found in the myeloid compartment but rather in the T cell compartment. Within NK cells, The NKp46+ cells in cluster largely lack CD16 expression and granzyme expression, which is odd. Cluster 2 looks to be underclustered, with an NKp46+ population towards the top of the tSNE that looks lower in GZMB, and then the remaining part of cluster 2 towards the bottom that largely lacks NKp46 expression but has high granzyme B. Overall, this raises questions about the nature of the clustering and assignment of cell identities. I would suggest first utilizing other clustering methods (other than tSNE) and demonstrating that the observed results are robust to choice of clustering method. Second, I would suggest additional details about how cluster identity was assigned.
- Figure 1c, the authors note that, descriptively, there appears to be more diverse immune

phenotypes in the visualization for T compared to N and P. Can the authors please quantify this?

- The reported results for figure 2 do not completely reflect what is shown in the figure. For instance, the authors broadly state that S2 tumors had lower pro-inflammatory or activated immune subsets (C2, C6, C9). But in the actual figure, The proportion of C2, for example is very similar in S1 and S2, and is only higher in S3. For C6, the proportion in S2 seems fairly similar to S3 this time, and it is lower in S1. Essentially, there is no common point of comparison – for some claims, S2 is higher or lower than compared only S1, and then for other claims, it is only higher or lower than compared to S3.
- For the manual gating in supplemental figure 2B, additional graphs would be helpful to further justify the claims. PD-1 is a marker of antigen experience and not just exhaustion. Further, exhausted T cells can still have expression of granzymes (this is common in terminally exhausted T cells). I would suggest the authors (1) provide additional support for calling the population exhausted (does it have expression of additional inhibitor checkpoints like LAG3 or TIGIT), and would also examine any differences in the PD1+GB+ cells across disease stages.
- For the scRNA-seq analysis, where samples all run at one time or in batches? If in batches, what were the batches and was batch correction performed. How was cluster identification performed? How many cells were used for each samples/patient, and does each cluster consist of cells from more than 1 patient? How many cells are in each cluster? These should be addressed in supplemental figures
- It is truthfully a bit unclear what the trajectory analysis in Figure 2e is trying to show. Typically, trajectory analysis is performed for a specific cell type to examine continuous changes in transcriptional state as one progresses along a biological process. I would suggest separating out cell types (CD8s, CD4s, etc.) and performing trajectory analysis on those individual, similar cell types.
- For figure 3d-e, it is a bit unusual that there is so little patient-to-patient variation in expression of individual genes within a stage. Usually there is heterogeneity between patients and at least some distribution, but here, there error bars seem incredibly small. As a minor point, I would not assume TCF7 as an exhaustion marker – while it can be associated with a progenitor exhausted state (which is typically favorable), it can also be associated simply with memory T cells
- For the neoantigen analysis, additional details would be helpful Was this truly whole genome sequencing, or whole exome? What protocol was used for sequencing? Further, the definition of clonal and subclonal are not typical. For instance, a subclonal mutation could be present in all biopsy sites, and would still be subclonal, not clonal.
- For figure 3g, it would be helpful to show a version of this exact plot but without PMA/ionomycin stimulation (i.e. showing the stim has an effect).
- Figure 4B is a bit of a confusing representation. Perhaps a heatmap or correlogram for these correlations would be easier to follow. As a small point, for figure 4C the top column labels are not aligned.
- For 4E, I would again caution against calling PD-1+ T cells exhausted, without any other indication, as fully functional antigen-experienced CD8+ T cells will also be PD-1+.
- In Supplementary Figure 4, the authors present in a heatmap the differentially expressed genes between samples across disease stages. This unannotated figure (no gene names) is not interpretable. It should be adjusted accordingly (the number of genes included in the heatmap could be reduced and/or samples from the same stage could be grouped).
- As a general but important point, there is not external validation of these findings, which would be important. A number of findings, for examples, are based on the expression of individual genes across disease stages. The authors could easily interrogate TCGA across disease stages to see if this validates.
- The discussion should be expanded to put this work in the context of other single-cell analyses of HCC.

Point-by-point response to reviewers' comments:

We are grateful for the reviewers' comments which help improve our manuscript tremendously. We have addressed their concerns with the point-by-point response below marked with **R:** and added these following major additional data:

1. We have expanded **scRNA seq data analysis (new Fig. 3a-c)**, with this additional new data, all the rest of the **figures numbers were shifted down** by one (***Fig. 3=> 4; Fig. 4=> 5; Fig. 5 => 6**).
2. We have provided more **detailed clinical analyses** as in point-by-point response to reviewers' comments below.
3. We have also added another **CyTOF clustering algorithm, FlowSom (new Suppl Fig. 2)** to show robustness of our data. Likewise the original Suppl Fig 4 to 6 have been shifted to Suppl Fig. 5 to 7.

All the other responses to each reviewer's comments are provided as point-by-point below:

REVIEWER COMMENTS

Reviewer #1 (Remarks to the Author):

In the paper by Nguyen P, et al, the authors comprehensively examined the immune subset changes in the different stages of HCC, by utilizing cutting-edge technologies, CyTOF, single cell RNA sequencing and exome sequencing. They found that the immunosuppression and exhaustion peaked in patients at stage II. The results of this research are of interesting and very important point to be addressed for future HCC therapy. However, some substantial issues are raised for the paper, especially in the study design and interpretation of the results.

Major points:

1. One of the major drawbacks of the study is the case analyses were performed only cross sectionally but not longitudinally. The authors categorized the patient group according to the TNM staging system. Although this classification is of clinical importance, patients background, such as tumor numbers, localizations, or vascular invasions, and liver function reserve, should be varied significantly even in patients in the same TNM stage. It is arguably interesting to see that the derangement of immune system peaked at the stage II but not at more advanced stage III. All the data provided by the authors, including a mouse model of chemical carcinogenesis (DEN-induced cancer), seemingly support for the results obtained from HCC patients. However, authors fail to provide some evidence, or at least some clues, to explain why such immune dysregulation peaked in the middle or intermediate stage of HCC patients. Do the authors think that CTNNB1 signature is a key? Are there any cases that the authors could examine the immune subset changes as shown here in some patients longitudinally, in the process of development from Stage I to II or III? And what are most significant factors, T, N or M, impacting on immune exhaustive features in patients?

R: Thank u for bringing up an excellent point, we believed the peak of immune evasion in intermediate stage of HCC is multifactorial, influenced by the tumour transcriptomic modification during tumour progression (Fig. 4), which impact and shape the tumour microenvironment as well as CTNNB1 upregulation and hence immune exclusion (Fig. 5) that concurrently push the TME towards exhaustion and evasion. We have added a sentence in discussion to discuss this view. *"This immune evasion could be due to the modifications in tumour transcriptomic landscape, which concurrently mirrored the immune microenvironment, with lower expression of genes involved in several immune-related pathways as well as upregulation of CTNNB1-related genes associated to immune exclusion, both occurred at S2 HCC tumours."*

As also stated in our manuscript in the discussion, “...*Similar to other human solid cancer studies, our current study has been limited to inferring the parallel cancer and immune evolution using specimens taken from different patients at various stages of tumour*^{11,35,36}...” to obtain longitudinal tumour samples from different stages of HCC is in fact virtually impossible. This is mainly because once the patients are detected with HCC tumours, they will be subjected to first-line therapy which is surgical resection if they meet the surgery criteria. There are no incidences, in fact even unethical, to biopsy the tumours and wait for tumour to progress to later stages to biopsy again. Even though some patients might not be eligible for surgical resection, they will definitely be given some sort of therapies such as locoregional therapies like radiotherapy or systemic therapy like Sorafenib (multiple tyrosine kinase inhibitor), which will and are known to significantly change the immune landscapes of the tumorus. For those cases, without surgery, taking biopsies are very rare and not recommended in fear of spreading the tumour in the process.

In view of these limitations, we have therefore performed **pseudotime trajectory analysis** on our scRNA seq data using monocle (Trapnell et al. Nat Biotech 2014), which uses an algorithm to learn the sequence of gene expression changes each cell must go through as part of a dynamic biological process. Once it has learned the overall "trajectory" of gene expression changes, Monocle can place each cell at its proper position in the trajectory and reconstruct a "branched" trajectory, which correspond to cellular "decisions". Monocle also provides powerful tools for identifying the genes affected by them and involved in making them. We have since expanded trajectory analysis to involve other immune cell types (**new Fig. 3**, see response to comments below).

In addition to single-cell pseudotime analysis, we also used the **murine model** where longitudinal study is possible to validate the continuous and progressive immune evasion along tumour progression.

We adopted version 8 TNM staging system (Kamarajah et al. J Surg Oncol. 2017;1–7) specified in method, for T, N or M which each represents T (primary tumour), N (regional lymph nodes) and M (Distant metastasis). According to the TNM staging v8 as illustrated in the table below, for all the stages I-III tumours involved in the current study, both N and M were not involved. Therefore, we are focusing primarily only on the T (primary tumour) status.

Table 1 extracted from Kamarajah et al. J Surg Oncol. 2017;1–7

[REDACTED]

2. The authors utilized DEN-induced carcinogenesis model as a backup for confirming the changes

of immune cell alterations according to the tumor progression. Because it is well acknowledged that DEN is a chemical carcinogen and induce liver tumors, the gene signature of which is completely different from that of human HCC. In addition, how the authors define the mice at 8month should be compatible with HCC patients at stage II? The authors need to explain the rationale for using DEN-model for the comparison.

R: We deeply understand the concerns from the reviewer on the compatibility of the genetic signature and progression of HCC between the DEN model and human patients. The genetic signature of DEN-induced HCC in mice has been extensively studied and compared to human HCC in the past two decades. Although DEN-induced HCC was initially found to closely recapitulate the advanced form of human HCC (Lee et al., Nat Genetics, 2004, 36:1306-1311), a more recent study revealed that other models such as STAM, MUP-uPA and TAK1 models could better recapitulate human HCC (Dow et al., PNAS, 2019, 115:E9879-E9888). Despite the differences in the genetic signature in the DEN and the other three mouse models, their immune cell infiltration, as indicated by the levels of mature CD4 T cells, T regulatory cells and dendritic cells, were found to be comparable between these mouse models, as well as to the human HCC (Dow et al., PNAS, 2018, 115:E9879-E9888). Furthermore, a previous study also demonstrated striking similarities in the regulation of T cell-associated genes between human HCC and the DEN model (Schneider et al., Gut, 2012, 61:1733–1743). In addition, DEN-induced HCC model went through a chronic inflammation process before HCC development where immunosurveillance has been demonstrated to suppress tumour development and progression much similar to human HCC (Schneider et al., Gut, 2012, 61:1733–43). Therefore, we reason that the DEN-induced tumors in mice provides a reliable model relevant to human HCC for investigating the changes in the immune landscape during HCC progression.

In fact, we did try another model which develops HCC model much faster within 21 days (typical endpoint) - **hydrodynamic tail vein induced HCC model** (Lin et al Cancer Res 2016). As a comparison (see figure below), even though indeed, level of PD-1+CD8+ T cells exhausted T cells are comparable between two models, active CD69+CD8+ T cells and Foxp3+CD25+CD4+ Treg are far lower for interrogating the changes in immune landscapes as intended. Given all considerations, we have decided to use DEN-induced HCC model despite the fact that we have to wait a longer duration of 6-12 months to complete the entire timeline experiment.

Figure: Mouse model comparison

For the comparison of stages between the DEN model and HCC patients, it is currently not possible to use the same TNM staging to define the tumor grades in the DEN model. In the DEN model, the tumor size increases progressively from 6, 8 to 12 months (**Fig. 5b**), this is indicative of tumour progression similar to human HCC. Prior to this study, whether the immune landscape undergoes remodelling during tumor progression in the DEN model remained largely unknown. When comparing the tumor size to the immune infiltration, activation and exhaustion within the tumors, we observed remarkable similarities between the 8-month DEN tumors and stage II human HCC in that the immune evasion appears to peak at these stages during the HCC progression.

Minor

1. In mice, it is important to see the efficacy of immune checkpoint inhibitors with/without Wnt inhibitors at different time points.

R: We have in fact planned this as the future work to test several immune checkpoint inhibitors and also to explore Wnt inhibitors in mice from different stages. However, this on its own involved a long time line (up to 12 months for advanced stages tumours in DEN-induced model) which will produce massive data. We therefore aim for it to be a separate study which we hope to complete in the near future. To acknowledge this important future study, we have included one sentence in the discussion to mention this.

“The efficacy of immune checkpoint inhibitors as well as Wnt inhibitors remain an interesting area for further research.”

2. The single cell sequence analysis in Figure2e may be useful to identify the downstream or upstream of S2 immune cell subset.

R: Indeed this is an excellent suggestion, we have since expanded Fig. 2e to a **new Fig.3** (also shown below) to explore other immune subsets from S1-S3 by single-cell analysis with the revised text in the manuscript as marked (also shown below).

New Fig. 3: Single-cell pseudotime trajectory analysis of immune cells along HCC progression

“Single-cell immune trajectory along tumour progression

To validate the above observation, we performed single-cell RNA sequencing on the tumour-infiltrating immune cells as previously described¹⁸ (**Supplementary Fig. 3a and Supplementary Table 5**) and analyzed their pseudotime trajectory along tumour progression using Monocle R package (Version 3.0), an algorithm to learn the sequence of gene expression changes each cell

*must go through as part of a dynamic biological process[19]. Setting S1 tumours as the starting point, we first explored the T cells trajectory along tumour progression S1-S3. Consistently, we observed a more distinct immune phenotype at S2 tumours with reduced CD69 and enriched PDCD1 (PD-1), FOXP3 and CTLA4 compared to N or S1 and S3 tumours (**Fig. 3a**). Consistent trend was also observed when analysed on CD4$^+$ T and CD8$^+$ T cells separately where Treg markers and exhaustion markers were found to be enriched in S2 tumours respectively; while proinflammatory gene GZMB were depleted in S2 tumours from both T cell subsets (**Supplementary Fig. 3b**). From the NK cell subsets, proinflammatory cytokines such as IFNG and TNF; chemokine, XCL1, which could promote cDC1 recruitment critical for antitumor immunity[20] as well as TNFSF14, which is important for antitumour function of NK cells via DC maturation[21]; were all depleted from S2 tumours but enriched in S1 or S3 tumours (**Fig. 3b**). In addition, we also explored mononuclear phagocytes (MNPs) populations with monocle pseudotime trajectory analysis (**Fig. 3c**) found several genes as enriched in S2 such as: SPP1, which has been shown highly expressed in tumour-associated macrophages (TAM) that promotes polarization to M2 phenotype facilitating immune escape[22] as well as TREM2, which marks TAM with highly immunosuppressive activity[23]. On the other hand, CD38, where its expression on myeloid cells predicts favourable prognosis in HCC hence suggesting its anti-tumoural acitvity[24] and proinflammatory cytokine TNF, were both depleted in S2 and S3 tumours.*
*Taken together, this data further corroborates a peak of phenotypic immune evasion in S2 HCC tumours."*

* Note: Due to the additional Fig.3, the manuscript now has a total of 6 figures. All the rest of the figures have been shifted down accordingly (***Fig. 3=> 4; Fig. 4=> 5; Fig. 5 => 6**).

3. Same as Figure4c, are there any significance on receptors for chemotaxis?

**R:** We have in fact included chemokine receptors in Fig. 4c (* **revised Fig. 5c**) such as *CCR6, XCR1, S1PR1, ACKR3*. We have now also **added few more** corresponding chemokines receptors for the chemokines presented in Fig. 4c: **CCR1** (receptor for CCL23 and CCL8), **CXCR1** (receptor for CXCL6), **CXCR2** (receptor for CXCL1, CXCL6) and **CXCR3** (receptor for CXCL9, CXCL10 & CXCL11). All of these chemokine receptors showed consistent trend with their corresponding chemokines (please see the revised Fig. 5c and also presented below with chemokine receptors marked as arrows). The text referring to result in Fig. 5c has also been revised as marked to describe this result.

*"Notably, the chemokines and their corresponding chemokine receptors: CCR1 (receptor for CCL23 and CCL8), CXCR2 (receptor for CXCL1, CXCL6) and CXCR3 (receptor for CXCL9, CXCL10 & CXCL11) showed consistent trend in correlation with the key immune subsets (Fig. 4c)."*

**Revised Fig. 5c**

Reviewer #2 (Remarks to the Author):

Nguyen et al characterized immune microenvironment in a series of 38 HCC with TNM stage I to III. They performed CyTOF analyses of tumor, non-tumor liver and blood samples for each patient. They identified immune depletion mainly in stage 2 patients. Immune evasion was validated in mice DEN model

Criteria of selection of the patients should be better described. In particular, in sup table 1 and in the text it is mentioned that patients with stage I, II or III were comparable. However, in sup table 1 it appears that stage II tumors were more aggressive with more MVI and proliferation. Why? Does patients with stage III are highly selected as treatable by surgery? This is a major point to correctly interpret the “immune recovery” observed in stage III patients.

R: As shown in Sup table 1 (table below), the significant difference in MVI lies in fact between stage I and stage II ($p < 0.0001$), stage I and stage III ($p = 0.0009$) but not between stage II and stage III ($p = 0.35$) (p values calculated using Fisher’s exact test, see graphs in figure below). In fact more advanced stages tumours are expected to have higher MVI incidence however no significant statistical difference was observed between S2 and S3. Therefore, we would rule out the possibility of selected S3 patients that contribute to our immune recovery data.

MVI	S1	S2	S3	
Y	0 (0.0%)	10 (83.3%)	6 (60.0%)	<0.0001****
N	16 (100.0%)	2 (16.7%)	4 (40.0%)	

Figure: S1 vs S2, S1 vs S3 and S2 vs S3 Fisher’s exact test

In the analyses, cirrhotic patients should be compared to non-cirrhotic one. Also, in sup table 1 F4 patients should be described separately.

R: We tried breaking down to detailed Fibrosis scoring for the patients from F0 to F4 and analyse them separately for each Fibrotic status and the p val remained insignificant using Chi-square test ($p = 0.7884$) when compared among three tumour stages:

Fibrotic status	S1	S2	S3	P val
F0	3	1	0	0.7884
F1	1	2	2	
F2	4	2	1	
F3	4	3	4	
F4	4	4	3	

We have also tried F0 vs F1-3 vs F4 as suggested. Likewise p val is also insignificant (p val = 0.6426):

Fibrotic status	S1	S2	S3	P val
F0	3	1	0	0.6426
F1-F3	9	7	7	
F4	4	4	3	

We hence concluded that fibrotic status of the liver does not affect our conclusion in the current study and provided the detailed **F0-F4 data** in the **revised suppl Table 1**.

Since the majority of the patients are HBV infected, the precise status of HBV infection should be included in the analysis. In particular antiviral therapy and viral infection activity should be described and correlated with the immune status.

R: In fact as shown in Suppl Table 1 (also shown below), viral status does not contribute significantly to different tumour stages (p val= 0.8979). In the current study, our focus is on the immune landscapes from different stages of tumours, as far as viral hepatitis is concerned, the distribution of viral vs non-viral cases are **comparable among three stages of HCC**, hence not likely to impact on the analysis. Furthermore, the separate effect of viral hepatitis and HCC immune microenvironment has already been reported in Lim & Lee et al. Gut 2019 (<https://gut.bmj.com/content/gutjnl/68/5/916.full.pdf>). We therefore concluded that viral hepatitis status does not impact on our current findings.

Viral status	S1	S2	S3	
Hep B	11 (68.8%)	8 (66.7%)	6 (60.0%)	0.8979
NV	5 (31.2%)	4 (33.3%)	4 (40.0%)	

Figure 1c: stage II and stage III peripheral typing seems exactly SIMILAR

R: Thank you very much for spotting this error. We apologize for the error made when copying and pasting the image. Previously we mistakenly pasted the P_S3 images twice and the S1 image was actually missing. We have since **corrected the mistake**, as can also be seen before and after correction, with the **headings** from the data generated by the software directly. We now confirmed that all data presented in the **revised Fig. 1c** is correct for P, T or N compartments and stages: S1, S2 or S3. Due to the small prints of the headings, we have removed the labels in the final revised fig. 1c after confirming they are all accurate.

Before correlation:

After correction in Revised Fig. 1c:

Neo-antigen load in stage II and III: are they significantly different at figure 3f?

R: The p val comparing stage II & III is $p = 0.9515$ due to the large standard deviation of S3 tumours. To make this clearer, we have included the p val in the **revised Fig 4f** (*please note that we have expanded the single-cell data analysis to new Fig.3 and hence all the rest of the figures have been shifted down by 1 to a total of 6 figures). Even though comparing S2 vs S3 neoantigen load is not significantly different but the trend is clear especially it is higher compared to earlier stage 1 tumours, which is also been shown in our earlier study Nguyen & Ma et al. Nat Com 2021, the accumulation of neoantigen corresponds to tumour progression and immune evasion.

Revised Fig. 4f

Sup figure 5 list: over and under-expressed genes in CTNNB1 mutated tumors are unusual compared to previous publications. Please explain why?

R: In fact, the CTNNB1 mutation genes are in line with previous publications, where up-regulation of Wnt signalling pathway or activation of the CTNNB1 pathway that has been linked with immune exclusion. For instance, Luke et al. Clin Cancer Res 2019 which concluded across multiple cancer types that “*Activation of tumor-intrinsic WNT/ β -catenin signaling is enriched in non-T-cell-inflamed tumors.*” <https://pubmed.ncbi.nlm.nih.gov/30635339/>. However, most of these previous studies examine **DNA mutation (such as CTNNB1, AXIN1, AXIN2, APC)** instead of transcriptomic gene signature as we have shown in the current study.

Hence for **HCC specific transcriptomic signature**, we referred to CTNNB1-signature specifically from HCC reported by Lachenmayer, A. et al. Clin Cancer Res 2012 (Genes shown in Supl Fig.6 * note with additional new suppl. 2 for FlowSom CyTOF analysis, this fig number has been shifted down). Our data in Fig. 5a showed upregulation of over-expressed genes and downregulation of under-expressed genes in CTNNB1 signature specifically at S2 tumours, which coincides with immune exclusion or immune “cold” TME from S2 HCC tumours (Fig. 5e,f).

Reviewer #3 (Remarks to the Author):

This manuscript seeks to characterize changes in immune cell populations and immune functions in hepatocellular carcinoma (HCC) across different disease stages. Using mass cytometry (CyTOF), the authors analyze immune clusters in HCC and normal liver samples, as well as peripheral blood. The authors report, stage 2 HCC exhibited lower levels of active NK cells and active memory CD8+ T-cells and higher levels of immunosuppressive CD8+ T-cells, T-reg and exhausted memory T-cells. Using IHC/IF, the authors identified the highest density of proportion of PD-1+ CD8+ T-cells in stage 2 tumors. Single-cell RNA sequencing of tumor samples with trajectory analysis was reported to show reduced CD69 expression and increased PD-1, FOXP3 and CTLA4 expression in stage 2 HCC as compared to other tumor stages. In bulk RNA sequencing and pathway analysis, stage 2 disease presented with the highest expression of genes related to immune exhaustion while the expression of genes related to immune response and antigen presentation was lower in advanced disease.

This manuscript potentially represents a relevant contribution to the literature as it may help to better understand the shifts occurring within the tumor immune microenvironment in HCC across disease stages. A “cold” TME is identified in stage 2 disease and partially validated through the different analyses performed, from mass cytometry and IHC, to bulk and single-cell RNA sequencing.

However, there are still many issues that would need to be addressed:

- Methods: more details needed. Examples include for enzymatic dissociation, how long was this performed for? Please include in methods additional details, such as catalog numbers for enzymes, timing of dissociation. What are the specific products used (catalog numbers for enzymes)?

R: The isolation of tissues-infiltrating immune cells was performed using **enzymatic dissociation** with **500ug/mL collagenase IV** (Thermo Fisher Scientific, Cat#: 17104019) and **50ug/mL DNase I** (Roche, Indianapolis, IN, Cat#: 4716728001) with the incubation of **30min in 37°C**. We have now provided all these information in revised Methods under “Patient samples” as marked.

- For patient, list each sample, and how many cells were included in the final analysis?

R: The enzymatic dissociation and the following processing and data acquisition for CyTOF yielded different number of cells which ranges from 10,000 to 200,000 cells. However for data analysis, we down-sampled the data consistently for EPIC analysis to **10,000 cells per sample**, the same process was described in our previous EPIC pipeline paper in *Yeo et al. Nat Biotech 2020*. This is now highlighted as “underlined” in Methods under “CyTOF”.

- Were any samples filtered out for QC reasons (i.e. low viable cell numbers)? If so, how many samples, and what were the criteria for this. This should be included in the methods.

R: Indeed, some of the sectors were excluded from analysis if the number of isolated cells were less than 100,000 of viable cells, as freezing, thawing and staining process for CyTOF will not likely yield sufficient high quality results for downstream CyTOF analysis. However, this does not affect our general analysis and conclusion as: 1. This is in fact a rare occasion, only 2 tumour sectors were excluded from the entire cohorts. 2. we obtained multiple sectors from each tumour, all of the tumours analyzed have at least two representative tumour sectors where the average values could well represent the immune landscapes of each tumour. To clarify this, we have revised the Methods to elaborate on this:

“Only samples with more than 100,000 cells after thawing will be subjected to downstream CyTOF analysis (note: two tumour sectors were excluded from the entire cohort and data was collected from at least two sectors of each tumour in the study.”

Also of note, since we did not exclude any data for analysis per se, rather they were excluded upstream when they did not pass QC for any downstream data acquisition and analysis hence the report summary remains as “no data are excluded from analysis”.

- What is the representation of each sample in each cluster? Are certain clusters patient specific (or compartment-specific)? Please include in the additional figures to address this, such as stacked bar graphs for each cluster by tissue origin and by patient origin.

R: We have also performed patient level analysis of the clusters (See figure below) and concluded that no particular clusters were patient specific and indeed they were more distinct when compared with tissue origin as already shown in Suppl Fig. 1b, where diversity is more apparent in Tumour (T) compartment. We have now included this pat level comparison UMAP in **revised Supl Fig. 1b**.

Revised Supl Fig. 1b

- The recovered immune population somehow appear off. First, compared to the work of Zhang et al, Cell, 2019, there appear to be many fewer myeloid cells. In reviewing the feature plots in the supplement, PD-L1 is not found in the myeloid compartment but rather in the T cell compartment. Within NK cells, The NKp46+ cells in cluster largely lack CD16 expression and granzyme expression, which is odd. Cluster 2 looks to be underclustered, with an NKp46+ population towards the top of the tSNE that looks lower in GZMB, and then the remaining part of cluster 2 towards the bottom that largely lacks NKp46 expression but has high granzyme B. Overall, this raises questions about the nature of the clustering and assignment of cell identities. I would suggest first utilizing other clustering methods (other than phonograph) and demonstrating that the observed results are robust to choice of clustering method. Second, I would suggest additional details about how cluster identity was assigned.

R: Indeed, it may seem that only few clusters are myeloid populations from our current study, this is because unlike the single-cell RNA seq data in Zhang et al Cell 2019, our current data is obtained from CyTOF analysis with defined surface and intracellular **protein markers** expression focusing more on T and NK cells but less on Myeloid cells (List of markers provided in Suppl Table 2). Furthermore, the current study analyze protein immune markers expression, which is more direct and crucial for the actual immune functions, compared to the RNA expression data shown in Zhang et al Cell 2019 previously.

Indeed, some of the markers expression are low such as PD-L1 and CD16, where the relative expression may appear in the “wrong” subsets but we relied on other more hallmark markers to define these clusters. For instance, APC, we used HLA-DR or in some cases CD14 expression for its definition. For NK, these subsets express the bona feta CD56 as well as CD244, well know NK

markers. Cluster 2 vs 16 are indeed two different NK populations, where with more clustering using FlowSom clustering (10x10= 100 clusters as shown below) could actually result in many more clusters but with similar phenotypes with only subtle differences between one another. Secondly, to address the reviewer's concerns for alternative clustering method, we have repeated the analysis using another well-established methods **FLowsOM** (10x10=100 clusters to provide the most detailed clustering) and showed **robust conclusion** as now included in **new suppl Fig. 2a-c**. The results section were also revised as "marked" to describe this additional supporting data.

"We also validated the above data using another clustering methods FlowSOM¹⁷ and found consistent trend of immune changes. Particularly, the immunosuppressive Treg and exhausted CD8+ T cells peaked in S2 tumours while the active CD8+ T cells and NK cells showed the lowest frequencies in S2 tumours (Supplementary Fig. 2a,b). Likewise, we validated the depletion of Treg in S2 PBMC and reduction of NKT cluster in S2 NILs (Supplementary Fig. 2c)."

However due to more clusters with similar phenotypes with FlowSom clusters, we still prefer to use the phonograph clustering. Furthermore, we went on to provide manual gating validation (**Suppl Fig. 2b => Fig. 2e *** in view of the importance of this data, we have move it to main fig 2e. Also to accommodate more scRNA seq data now expanded to new Fig. 3) to support the trends in these clusters with specific immune subsets.

New Supl Fig. 2: HCC immune landscapes analysis from different stages using FlowSom algorithm

a

Trends in Tumour

Highest at S2

**

*

Lowest at S2

**

*

APC

B

NKT

NK

CD8+ T

CD4+ T

Treg

PD-1-GzmB+
CD8+ T cells

PD-1+CD8+
exhausted T cells

GzmB+ active
NK cells

CD69+ active
NK cells

b

C

- Figure 1c, the authors note that, descriptively, there appears to be more diverse immune phenotypes in the visualization for T compared to N and P. Can the authors please quantify this?

R: Indeed, Fig. 1c was meant only to provide a global landscape presentation of immune changes among different stages of tumour. The increased diversity of immune phenotypes was also demonstrated in **Suppl Fig. 1b**. In addition, the quantification of clusters with significant changes were provided in **Fig. 2b-d** (shown below for reference), where indeed more clusters showed significant changes in T compartment compared to N & P. The similar findings were found also in 10x10 FlowSom analysis provided in **new Suppl Fig. 2a-c** as described above.

Suppl Fig. 1b: Increased immune diversity in T

Fig.2b-d

- The reported results for figure 2 do not completely reflect what is shown in the figure. For instance, the authors broadly state that S2 tumors had lower pro-inflammatory or activated immune subsets (C2, C6, C9). But in the actual figure, The proportion of C2, for example is very similar in S1 and S2, and is only higher in S3. For C6, the proportion in S2 seems fairly similar to S3 this time, and it is lower in S1. Essentially, there is no common point of comparison – for some claims, S2 is higher or lower than compared only S1, and then for other claims, it is only higher or lower than compared to S3.

R: Indeed, the comparison of these clusters among three stages was meant to show the general trend where the most differences occurred between S2 with either S1 or S3 tumours. To avoid confusion, we have now added more detailed description with specific p values and comparison pairs for each cluster when describing fig. 2b-d in the revised manuscript.

“The tumour tissue exhibited the greatest inter-stage heterogeneity...with the majority of these showing significant differences at S2 (Fig. 2a). S2 tumours exhibited lower levels of pro-inflammatory or activated immune subsets including GB+CD16+CD56+ active NK cells (C2, $p=0.0471$, S2 vs S3), PD-1-GB+CD45RO+ (C6, $p=0.0220$, S1 vs S2), and PD-1-CD69+CD45RO+ (C9, $p=0.009$, S1 vs S2; $p=0.0184$, S2 vs S3) active memory CD8+ T cells (Fig. 2d). Conversely, the frequencies of exhausted and immunosuppressive clusters PD-1+GBlowCD45RO+CD8+ T cells (C1, $p=0.0458$, S2 vs S3), Treg (C4, $p=0.0188$, S1 vs S2), and PD-1+GBlowCD45RO+CD4+ exhausted memory T cells (C19, $p=0.0073$, S2 vs S3) were significantly higher in S2 tumours than in either S1 or S3 (Fig. 2d).”

More importantly, the conclusion of significant immune evasion peaking at S2 was supported by manual gating of key immune subsets in suppl Fig. 2b. In view of this important data, we have moved this to main Fig. 2e.

Suppl Fig. 2b => Fig. 2e

- For the manual gating in supplemental figure 2B, additional graphs would be helpful to further justify the claims. PD-1 is a marker of antigen experience and not just exhaustion. Further, exhausted T cells can still have expression of granzymes (this is common in terminally exhausted T cells). I would suggest the authors (1) provide additional support for calling the population exhausted (does it have expression of additional inhibitor checkpoints like LAG3 or TIGIT), and would also examine any differences in the PD1+GB+ cells across disease stages.

R: Indeed, PD-1 could also be a sign of antigen exposure and may not necessarily mean exhaustion, therefore, to show its exhausted phenotype we did gate for granzyme B and showed these cells express low GzmB (Median 21.85% ± 23.05%, **new suppl fig. 3b-d** and shown below for reference). Also, we looked into other exhaustion markers including CTLA-4, Tim-3 and Lag3 and found that these PD-1+CD8+ T cells indeed co-expressed a number of exhaustion markers (shown below): CTLA-4 (Median 66.2% ± 21.71%), Tim3 (Median= 72.4% ± 17.61%) and to a lesser extent Lag3

(Median $36.3\% \pm 17.07\%$), again supporting its exhausted phenotypes (**new suppl Fig.3b-d**). Also, as shown by our previous paper in Chew et al. PNAS 2017, we have done comprehensive analysis of PD-1+CD8+ T cells and showed their exhausted phenotypes in HCC TME. We added one sentence *“marked”* in the revised result section to describe this new data.

“Of note, the PD-1+CD8+ T cells expressed very low level of granzyme B (Median $21.85\% \pm 23.05\%$) and co-expressed high level of CTLA-4 (Median $66.2\% \pm 21.71\%$), Tim3 (Median= $72.4\% \pm 17.61\%$) and to a lesser extent Lag3 (Median $36.3\% \pm 17.07\%$) (Supplementary Fig. 3c-e), supporting its exhausted phenotypes as also shown by our previous study of exhausted PD-1+CD8+ T cells in HCC¹⁸.”

New Suppl Fig. 3b-d

- For the scRNA-seq analysis, were samples all run at one time or in batches? If in batches, what were the batches and was batch correction performed. How was cluster identification performed? How many cells were used for each samples/patient, and does each cluster consist of cells from more than 1 patient? How many cells are in each cluster? These should be addressed in supplemental figures

R: Indeed, the scRNA-seq has to be run fresh after isolation hence each patient data was acquired in batches and all data were then aggregated using cellranger aggr by normalizing all runs to the same sequencing depth as also described in our collaborator’s previous publication in Sharma et al. Cell 2020. For clustering, best matched k-Nearest Neighbor is automatically weighted by the algorithm to compute the best UMAP topology (scanpy.api.tl.umap, minimum distance between 0.3 to 0.5). The Louvain method (scanpy.api.tl.louvain) is then used to detect a community of similar cells with resolution parameter 0.6 to 1. This detailed method is now added to revised Methods section under *“Single-Cell RNA sequencing”* as marked. Each of the T, NK and MNPs clusters with total cell numbers as well as proportions contributed by each patient sample are now also provided in **Supplementary Fig. 3a and Supplementary Table 5**.

Revised Suppl Fig. 3a

[REDACTED]

- It is truthfully a bit unclear what the trajectory analysis in Figure 2e is trying to show. Typically, trajectory analysis is performed for a specific cell type to examine continuous changes in transcriptional state as one progresses along a biological process. I would suggest separating out cell types (CD8s, CD4s, etc.) and performing trajectory analysis on those individual, similar cell types.

R: Pseudotime trajectory analysis in Fig. 2e (now **new Fig. 3**) is based on scRNA seq data analysis using monocle (Trapnell et al. Nat Biotech 2014), which is an algorithm to learn the sequence of gene expression changes each cell must go through as part of a dynamic biological process. Once it has learned the overall "trajectory" of gene expression changes, Monocle can place each cell at its proper position in the trajectory and reconstruct a "branched" trajectory, which correspond to cellular "decisions". Monocle also provides powerful tools for identifying the genes affected by them and involved in making them. To clarify this, we have also added more description in the results referring to monocle analysis as "*marked*" in the revised manuscript. More information is also available online: <https://cole-trapnell-lab.github.io/monocle3/docs/introduction/>
For T cell trajectory analysis, we also tried performing the trajectory analysis separately for CD4+ (due to low CD4 gene expression, we used CD8-CD3+ as CD4+ T cells) and CD8+ T cells (CD8+CD3+ T cells) and concluded consistent and robust findings as also marked in revised manuscript: "*Consistent trend was also observed when analysed separately in CD4+ T and CD8+ T cells where Treg markers and exhaustion markers were enriched in S2 tumours respectively; while proinflammatory gene GZMB were depleted in S2 tumours from both T cell subsets (Supplementary Fig. 3b).*"

New Suppl Fig. 4b

- For figure 3d-e, it is a bit unusual that there is so little patient-to-patient variation in expression of individual genes within a stage. Usually there is heterogeneity between patients and at least some distribution, but here, there error bars seem incredibly small. As a minor point, I would not assume TCF7 as an exhaustion marker – while it can be associated with a progenitor exhausted state (which is typically favorable), it can also be associated simply with memory T cells

R: We chose to present the data as Mean±SEM as this does provide better representation of the data where SD would have bigger error bars that may block the trend of other genes when all were presented together on the same graph (see the graphs below presented as SD). We did specify that data are presented as this in the figure legend that we have presented the data as “mean with standard error of the mean” as we do find this presentation as neater and clearer. Furthermore all the pairwise comparison p values were provided on top of the graph to support the significant differences comparing among groups in colours.

Original Fig. 3d & 3e as presented as mean ±SD:

As for TCF7, indeed, recent studies have indicated that TCF7 is found to be expressed on the stem like memory T cells involved in better response to immunotherapy (e.g. Siddiqui et al., 2019-ref 21 & Immunity or Krishna et al Science 2020-ref 22). However, another Immunity paper in Chen et al 2019 (ref 20) did also show that TCF1 plays an central role in establishing exhausted T cells: [https://www.cell.com/immunity/pdf/S1074-7613\(19\)30409-1.pdf](https://www.cell.com/immunity/pdf/S1074-7613(19)30409-1.pdf)

To illustrate these point better, we have revised the manuscript to describe both its potential stem-like memory phenotypes as well as exhausted phenotype in the revised manuscript as marked.

“Of note, TCF7 despite being a key factor in establishing exhausted T cells²⁰, could also present a stem-like memory T cell phenotype associated to response to immunotherapy^{21,22}.”

- For the neoantigen analysis, additional details would be helpful Was this truly whole genome sequencing, or whole exome? What protocol was used for sequencing? Further, the definition of clonal and subclonal are not typical. For instance, a subclonal mutation could be present in all biopsy sites, and would still be subclonal, not clonal.

R: Yes, we did use whole genome sequencing (WGS) with details provided in methods “*Whole genome sequencing and neoantigen prediction*”. It is also the same methods we used in our previous paper in Nat Com Nguyen & Ma et al. 2021 (<https://www.nature.com/articles/s41467-020-20171-7>). We have now mentioned this in the text (under Results section) when describing neoantigen data:

“We then quantified neoantigens in tumour samples from each stage using the WGS data (Methods) and found...”

We took the definition of clonal and subclonal neoantigens from McGranahan et. al (Science 2016) (<https://science.sciencemag.org/content/351/6280/1463>), which has defined clonal neoantigens as expressed ubiquitously in every tumor region and subclonal neoantigens as shared in multiple tumor regions but not all. To specify this, we have now cited this paper in Methods to provide reference to our definition.

- For figure 3g, it would be helpful to show a version of this exact plot but without PMA/ionomycin stimulation (i.e. showing the stim has an effect).

R: We did not include this data before as without stimulation with PMA/Ionomycin, the level of cytokines are very low in fact almost all are 0% across all samples (representative dot plots now included in **new suppl fig. 3e** for reference). As a control, we have now included unstimulated data at the side for comparison in the **revised Fig. 4g** (*note the shift from Fig. 3g to Fig. 4g).

New Suppl Fig. 3e

Revised Fig. 4g:

- Figure 4B is a bit of a confusing representation. Perhaps a heatmap or correlogram for these correlations would be easier to follow. As a small point, for figure 4C the top column labels are not aligned.

R: Indeed, Fig. 4B (now shifted to Fig. 5b) was meant to provide a visual correlation landscapes of some of the key genes of interest with different immune subsets, which we later elaborated further with further selected key genes using heatmap and correlogram in Fig 5c. In fact Fig. 4b does provide valuable general correlation landscapes when most negative correlations were with CD4 T cells but positive correlations were with CD8 T cells subsets.

- For 4E, I would again caution against calling PD-1+ T cells exhausted, without any other indication, as fully functional antigen-experienced CD8+ T cells will also be PD-1+.

R: We agree that there is a possibility that PD-1+ T cells could still be functional as newly activated by tumour antigen. However, in this case, these are TILs within TME, which are most likely chronically exposed tumour antigens and hence more likely to be exhausted. Also resonating our data from manual gating as shown above (**New Suppl Fig. 3b-d**), we would hence like to think they are exhausted.

- In Supplementary Figure 4, the authors present in a heatmap the differentially expressed genes between samples across disease stages. This unannotated figure (no gene names) is not interpretable. It should be adjusted accordingly (the number of genes included in the heatmap could be reduced and/or samples from the same stage could be grouped).

R: We agree that these genes list are too large for presentation visually hence we have in fact selected few key genes and presented them in **Fig. 4d, 4e, 5c & 5d** (*Note: Fig number have been shifted). However, we also think that it is important to provide a global picture of all the differentially expressed genes (DEGs) when compared among stages (**now as Suppl. Fig. 5a**). To allow better understanding of these DEGs, we have also provided the list of genes significantly associated with several functional pathways in **Suppl Table 6 and Suppl Table 7**, as part of the requirements from Nature Comm to report the global data as much as possible.

- As a general but important point, there is not external validation of these findings, which would be important. A number of findings, for examples, are based on the expression of individual genes

across disease stages. The authors could easily interrogate TCGA across disease stages to see if this validates.

R: We thank the reviewer for this suggestion and have interrogated the TCGA HCC cohort across S1, S2 & S3. In general as consistent to Fig. 4d, 4e, 5d, we observed a progressive downtrend of multiple key genes involved in antigen presentation, inflammatory response, exhaustion and CD8 T chemotaxis (see figure below), however with a more modest differences among stages. We believe the differences seen here may due mainly to a **different background** of these HCC patients in TCGA cohort, who may receive different prior therapies, such as radiotherapy and systemic therapy, which will change the immune landscapes. Our cohort of HCC patients however are **treatment naïve** hence showing the natural progression of tumour. In view of this, we have decided not to include this data to the revised manuscript.

Figure: TCGA interrogation of key genes in manuscript:

- The discussion should be expanded to put this work in the context of other single-cell analyses of HCC.

R: Thank you for the suggestion and we have now expanded the discussion to also comment on other recent single-cell analyses in HCC most relevant to our current study such as Zheng et al Cell 2017 (<https://doi.org/10.1016/j.cell.2017.05.035>), Zhang et al 2019 and the study from our own liver cancer team, Sharma et al. Cell 2020. We added the following sentences as marked in the discussion section:

“Immune evolution in HCC was proposed in our recent study to be manifested as immune intratumoural heterogeneity (immune-ITH)¹⁴. In addition, single-cell transcriptomic analysis of TILs in HCC also revealed immune evasion with T cell exhaustion³⁸ (Zheng et al Cell 2017), myeloid/lymphoid cross-interaction (Zhang et al 2019) and onco-fetal reprogramming of endothelial cells and macrophages¹⁷ (Sharma et al Cell 2020). In this study, we revealed an unprecedented progressive immune evolution alongside tumour progression peaking at the intermediate stage of HCC.”

REVIEWER COMMENTS

Reviewer #1 (Remarks to the Author):

The authors added several analytical data and responded mostly to the comments by the reviewer. However, the major points regarding to the peak immune derangement at intermediate HCC stage still remain ambiguous. It is understandable that longitudinal sampling of HCC tissues from patients is clinically impossible. Admittedly, the data from DEN or hydrodynamic mice model did not well support for the patients data. Only minor issues are raised as below.

Minor points:

It is reported that the mutation of Wnt signaling molecules including CTNNB1 and Axin1 correlates with not only immunologically cold phenotype, but also with pathologically large and well-differentiated HCC. The pathological phenotype in the patients list would be more helpful to understand the background of the donors.

Reviewer #2 (Remarks to the Author):

I have no additional comments. In my point of view, this paper include a huge amount of data, however, the peak of immune evasion at stage II remains to be validated and elucidated since the number of patients is limited

Reviewer #3 (Remarks to the Author):

The authors have revised the manuscript, and adjustments were performed in a number of instances. Clear explanations were provided for the different steps in the study (e.g. QC metrics, CyTOF, scRNA-seq), and corrections were applied to the different sections.

However, some substantial issues remain to be further addressed:

(1) For the CyTOF analysis: the authors were asked to assess if the identified clusters are patient-specific and compartment-specific. A proposed visualization method was clearly specified for how to assess this ("stacked bar graphs for each cluster by tissue origin and by patient origin"). The authors responded to this by only adding two UMAP plots (one for tissue of origin and one for patients) that do not help assess if any cluster is patient- or compartment-specific.

(2) The authors were asked to quantify immune diversity in tumor samples (as opposed to non-tumor and blood samples), to substantiate their statement ("the presence of more distinct immune clusters in T and to a lesser extent in N and P across all stages"). Supplementary Figure 1B (a UMAP plot showing distributions of immune clusters according to tissue level) helps to show an increased heterogeneity in tumor (T) tissues compared non-tumor (N) and blood (P) tissues, but does not help to quantify the immune diversity. Moreover, Figure 2b-d shows changes in clusters within individual tissue types for different stages of disease but does not help to compare or quantify immune diversity across sample types.

(3) When comparing clusters across different disease stages, no common point of comparison was found across the different clusters, with S2 being compared to S1 in some clusters and to S3 in other clusters. The authors didn't address this point clearly, as only a specification was added in the text for the nature of each comparison performed (i.e. the control stage vs. S2). While the results of the manual gating of key immune subsets helps to partially address this problem, the previously

presented results remain to be further adjusted, with the potential removal of some of the findings.

(4) For the scRNA-seq analysis, the authors were asked to specify the number of cells used for each samples/patient and each cluster, and to state whether each cluster consisted of cells from more of one patient. Supplementary Figure 3A shows the distribution of clusters in each patient. However, the inverse should have been evaluated (i.e. the contribution of patients to each cluster), hence helping to appreciate whether some clusters were driven by few patients. Additionally, the authors indicate that “each of the T, NK and MNPs clusters with total cell numbers as well as proportions contributed by each patient sample” are provided in Supplementary Table 5. However, Supplementary Table 5 displays the “Top 25 genes for T, NK and myeloid/DC clusters by scRNA sequencing analysis (adapted from Sharma et al. Cell 2020)”, and the specified numbers/proportions are not found in any of the Supplementary Tables. This should be adjusted accordingly.

(5) The definitions of clonal and subclonal mutations presented by the authors are not typical. Importantly, a mutation can be present in all biopsy sites (or tumor regions) and still be subclonal. The authors didn't address this issue appropriately and cited the paper by McGranahan N. et al. (Science, 2016) to justify their definition of subclonality and clonality. However, it is clearly stated in the same paper (referenced by the authors in their response) that clonal mutations are “present in all tumor cells” (as opposed to all biopsy sites of a tumor), while subclonal mutations are “present in only a subset [of cells]”. A simple and clear example of this would be a mutation that is present in only 10% of tumor cells, but is found in 10% of tumor cells in all biopsy sites / tumor regions. This is still, by definition, a SUBCLONAL mutation (found in only 10% of tumor cells), even though it might be seen in all tumor regions.

(6) In their response, the authors argue that PD-1+ T-cells represent most likely an exhausted population as “these are TILs within TME, which are most likely chronically exposed tumor antigens”. This statement is not completely correct, as it has been previously shown that CD8+ PD-1+ tumor-infiltrating cells correlate with high levels of T-cell activation and are associated with better outcomes (Pignon J.C. et al., Clinical Cancer Research, 2018; PMID: 30670497). While the analysis of immune checkpoint markers (i.e. LAG-3, TIM-3) on PD-1+ cells favors the exhausted nature of these cells, the first statement should be clarified.

(7) The authors were asked to externally validate their findings using publicly available datasets, such as the TCGA. The findings do not appear to validate in the TCGA. The authors note that this “may due mainly to a different background of these HCC patients in TCGA cohort, who may receive different prior therapies, such as radiotherapy and systemic therapy, which will change the immune landscapes”. I do not believe this is correct – the sample inclusion criteria for HCC for TCGA (from the original 2017 Cell manuscript) specifically states it is patients who had not received prior treatment for their disease. With this in mind, it is concerning that the findings do not appear to validate in an external dataset.

Point-by-point response to reviewers' comments:

Reviewer #1 (Remarks to the Author):

The authors added several analytical data and responded mostly to the comments by the reviewer. However, the major points regarding to the peak immune derangement at intermediate HCC stage still remain ambiguous. It is understandable that longitudinal sampling of HCC tissues from patients is clinically impossible. Admittedly, the data from DEN or hydrodynamic mice model did not well support for the patients data. Only minor issues are raised as below.

Minor points:

It is reported that the mutation of Wnt signaling molecules including CTNNB1 and Axin1 correlates with not only immunologically cold phenotype, but also with pathologically large and well-differentiated HCC. The pathological phenotype in the patients list would be more helpful to understand the background of the donors.

R: We thank the reviewer for the additional suggestion to explore if Wnt signalling is also linked to pathological phenotypes: tumour size and tumour differentiation. Herein, we provide the analysis in the fig below based on correlation of CTNNB1 signature with tumour Edmondson Grade, which reflects tumour differentiation (**Fig. a**) and tumour size (**Fig.b**). We concluded that in our dataset, they are not significantly correlated to tumour differentiation or size.

As the demographic characteristics of the patients are already summarized in Supplementary Table 1. Herein, we provide the pathological phenotypes of our patient cohort in the list below for your kind reference:

Patient demographic characteristics:

Pat ID	Age	Gender	Race	Stage (TNM)	Grade (Edmondson)	Viral status	Tumour size (cm)	Tumour multiplicity	AFP level (ng/ml)	MVI
A001	66	M	Chinese	I	IV	NV	2	1	6.7	N
A002	74	M	Chinese	I	II	Hep B	3.5	1	2.8	N
A004	77	M	Chinese	II	II	Hep B	3	1	38	Y
A005	65	M	Chinese	I	III	Hep B	1.8	1	69.3	N
A008	70	F	Malay	IB	II	NV	3.8	1	248	N
A009	47	F	Chinese	II	III	NV	7.5	1	2203	Y
B002	52	M	Chinese	I	II	Hep B	2.5	1	4.7	N
B003	60	M	Chinese	IIIC	III	NV	11	1	1.9	Y

B004	52	M	Chinese	I	III	Hep B	2.9	1	13.5	N
B006	67	F	Chinese	I	III	Hep B	3.6	1	8592	N
B008	78	M	Chinese	IIIC	III	Hep B	7	1	36.6	Y
B009	67	M	Malay	IIIC	III	Hep B	9.5	1	>60500	Y
B010	74	F	Chinese	I	I	NV	2.5	1	5.4	N
B012	62	M	Chinese	I	I	Hep B	2.6	1	4.2	N
B013	57	M	Chinese	IIIA	II	Hep B	8	multifocal satellite	49.9	N
B014	73	M	Chinese	II	II	Hep B	14	1	1.5	Y
B015	63	M	Chinese	IB	II	Hep B	2.2	1	6.1	N
B016	71	M	Chinese	IB	II	Hep B	4.7	1	8920	N
B017	75	M	Chinese	II	III	Hep B	6.2	1	504	Y
B018	46	M	Chinese	I	III	Hep B	4.7	1	657	N
B019	63	M	Chinese	III	II	Hep B	5.5	2	62	Y
C002	69	M	Chinese	IIIA	II	NV	12.7	2	1069	N
C003	72	M	Chinese	II	II	Hep B	3.5	2	3	N
C004	69	F	Chinese	I	II	Hep B	8.5	1	999.99	N
C005	61	M	Others	IIIB	III	NV	8	1	144.1	N
C008	75	M	Chinese	IIIA	II	Hep B	14	Satellite nodule	13.7	N
C010	55	M	Indian	IB	II	NV	15	1	9.3	N
C011	70	M	Chinese	II	II	Hep B	4	Satellite nodules	6.5	N
C012	71	M	Chinese	II	III	NV	14	1	54458	Y
C014	60	M	Chinese	II	II	Hep B	12.2	1	2.7	Y
C015	75	M	Chinese	II	II	Hep B	11.2	1	2.8	Y
H156	66	M	Chinese	I	III	Hep B	3	1	4.3	N
H247	82	F	Chinese	II	III	NV	3.5	1	27.3	Y
H255	71	F	Chinese	II	III	Hep B	5	1	>10000	Y
H264	64	M	Malay	II	II	NV	8	1	20.1	Y
H276	76	M	Chinese	I	III	NV	5.5	1	3.2	N
H319	69	M	Indonesian	IIIC	III	NV	5.6	1	14483	Y
H526	43	M	Others	III	II	Hep B	6.5	2	2985	Y

Footnotes:

Gender: M- Male; F-Female

Viral status: HBV- Hepatitis B virus; NV- non-viral

AFP: Alpha-Fetoprotein

MVI: microvascular Invasion; Y- Yes; N- No

Reviewer #2 (Remarks to the Author):

I have no additional comments. In my point of view, this paper include a huge amount of data, however, the peak of immune evasion at stage II remains to be validated and elucidated since the number of patients is limited

R: We acknowledge this point and are actively recruiting more patients for future validation under our observational clinical trial: <https://clinicaltrials.gov/ct2/show/NCT03267641>

Reviewer #3 (Remarks to the Author):

The authors have revised the manuscript, and adjustments were performed in a number of instances. Clear explanations were provided for the different steps in the study (e.g. QC metrics, CyTOF, scRNA-seq), and corrections were applied to the different sections.

However, some substantial issues remain to be further addressed:

R: We appreciate the reviewer for the thorough assessment and further comments to help us improve our manuscript. We are happy to provide further supporting data and clearer explanation to address each comment as below.

(1) For the CyTOF analysis: the authors were asked to assess if the identified clusters are patient-specific and compartment-specific. A proposed visualization method was clearly specified for how to assess this (“stacked bar graphs for each cluster by tissue origin and by patient origin”). The authors responded to this by only adding two UMAP plots (one for tissue of origin and one for patients) that do not help assess if any cluster is patient- or compartment-specific.

R: It is definitely possible to provide the data as stacked or colour graded bar graphs (see below) even though we proposed to present the same data using UMAP plots as the differences between tissues and stages can be appreciated more easily with this visual representation. We would propose to show this bar graph in **new Suppl. Fig. 1a (as shown below)** where the cluster frequencies, represented by colour gradient, for each patient are plotted in each column. As expected, there are definitely some smaller clusters which have lower frequencies across samples, despite that the clusters frequencies were subjected to statistical test with all significant clusters as reported in **Fig. 2b**, where the individual data point was also plotted supporting the robustness and statistical significance of our data.

New Suppl. Fig. 1a:

(2) The authors were asked to quantify immune diversity in tumor samples (as opposed to non-tumor and blood samples), to substantiate their statement (“the presence of more distinct immune clusters in T and to a lesser extent in N and P across all stages”). Supplementary Figure 1B (a UMAP plot showing distributions of immune clusters according to tissue level) helps to show an increased heterogeneity in tumor (T) tissues compared non-tumor (N) and blood (P) tissues, but does not help to quantify the immune diversity. Moreover, Figure 2b-d shows changes in clusters within individual tissue types for different stages of disease but does not help to compare or quantify immune diversity across sample types.

R: Suppl Fig. 1B (now **suppl Fig. 1c** due to additional data added) in fact does provide indication of immune diversity/heterogeneity as the fundamental principle of UMAP plot is to show distribution of (immune) clusters according to their phenotypic similarity whereby the closer the immune clusters are clustered together, the more similar or homogeneous (e.g. P & N) and the further apart means more heterogeneous (T) they are. However, as requested by the reviewer, we also specifically **quantified the immune diversity**. Heterogeneity of immune cell phenotypes was quantified by calculating the **multivariate beta-dispersion of the Bray-Curtis distances between immune clusters** for each tissue type using the ‘vegan’ R package. The beta-dispersion were compared among tissues and p values were determined using Tukey’s HSD (honestly significant difference) test.

As shown below and now included as **new suppl. Fig. 1d**, we observed significant diversity (enhanced distance) between immune clusters comparing T vs N ($p=0.0012^{**}$) and T vs P ($p=0.0041^{**}$) while no significant difference when comparing P vs N ($p=0.6997$).

New Suppl. Fig. 1d

Tissues	Difference	P adj.
P vs N	0.01344680	0.6997266
T vs N	0.05640242	0.0001230
T vs P	0.04295562	0.0040921

With regards to the second request “to compare or quantify immune diversity **across sample types**”, we have in fact performed the comparison across tissue types at different tumour stages before however we did not show this data as we were concerned there might be too much data and confusing for the readers. Furthermore, we found that the immune differences are heavily skewed by tissue types where across all three tumour stages where we see consistently 13 immune clusters showing the apparent more immune exhaustion in T as compared to P or N, a conclusion which has already been reported by many previous studies including our own earlier study in Chew et al PNAS 2017.:

Despite all that, as requested we have performed statistical analysis comparing P vs N vs T (across sample types) at each individual stage I, II & III and now present the full data in **new suppl Table 3 and suppl Fig. 2** with the descriptions in the manuscript text provided as below:

“From all three tumour stages, 13 cluster showed significant differences in their frequencies across tissue types: P, N and T using One-way Anova test (Supplementary Fig. 2a and Supplementary Table 3). Among these, we observed enrichment of potentially exhausted and suppressive immune subsets in tumours: $Foxp3^+CD152^+TIGIT^+CD4^+$ regulatory T cells (Treg) (C4), $PD1^+CD103^+CD45RO^+CD8^+$ resident memory T cells (TRM) (C7) and $PD1^+CD45RO^+CD4^+$ memory T cells (C19) (Supplementary Fig. 2b). Conversely, the frequencies of immunoactive subsets including GB^+CD56^+ NK cells (C2) and $GB^+CD56^+CD8^+$ NKT cells (C18) were significantly lower in T than N or P (Supplementary Fig. 2c). Moreover, we saw $PD-1^+GB^{lo}CD45RO^+CD8^+$ memory T (C1), $CD69^+CD8^+$ memory T cells (C9) and $PD-L1^+CD45RO^+CD4^+$ memory T cells (C13) were enriched in both N and T compared to P (Supplementary Fig. 2d); while the antigen-presenting $HLA-DR^+CD19^+$ B cells (C8) and

HLA-DR⁺CD14⁺ myeloid cells (C10), as well as CD27⁺CD45RO⁻CD4⁺ naïve T cells (C17) were significantly reduced in both N and T compared to P (Supplementary Fig. 2c). This shows a general accumulation of memory subsets and depletion of antigen-presenting cells in the N and T compartments as compared to P. Together, these data indicate that the immune evasion is established early and maintained throughout the following stages of tumor development.”

New suppl. Table 3:

Supplementary Table 3: 13 clusters showing significant differences comparing P, N and T at different tumour stages

Stage I:

Cluster= C	pvalOwa		p_MWU_P_S1 vs N_S1		p_MWU_P_S1 vs T_S1		p_MWU_N_S1 vs T_S1		Median_P_S1	Median_N_S1	Median_T_S1
0	0.0024	**	0.7283		0.0107	*	0.0020	**	6.460	5.304	10.924
1	0.0025	**	0.0001	***	0.0000	***	0.2261		1.320	9.767	7.862
2	0.0021	**	0.4059		0.0025	**	0.0467	*	10.560	10.300	3.641
4	0.0000	***	0.0135	*	0.0000	***	0.0000	***	1.860	0.880	6.343
5	0.0003	***	0.0001	***	0.0001	***	0.0073	**	0.920	10.985	3.820
7	0.0001	***	0.0015	**	0.0000	***	0.7777		0.400	3.462	4.400
8	0.0061	**	0.0066	**	0.0052	**	0.7679		5.762	2.440	2.001
9	0.0000	***	0.0004	***	0.0000	***	0.1640		0.700	4.820	2.961
10	0.0000	***	0.0160	*	0.0008	***	0.2985		7.640	1.190	0.900
13	0.0346	*	0.0868	.	0.0078	**	0.3422		1.260	1.970	2.760
17	0.0000	***	0.0003	***	0.0000	***	0.5549		10.746	0.500	0.400
18	0.0201	*	0.4371		0.0043	**	0.0703	.	1.481	1.370	0.940
19	0.0000	***	0.0682	.	0.0000	***	0.0004	***	0.200	0.780	2.260

Stage II:

Cluster= C	pvalOwa		p_MWU_P_S2 vs N_S2		p_MWU_P_S2 vs T_S2		p_MWU_N_S2 vs T_S2		Median_P_S2	Median_N_S2	Median_T_S2
0	0.0000	***	0.7987		0.0000	***	0.0000	***	6.480	6.291	14.633
1	0.0002	***	0.0000	***	0.0000	***	0.6177		0.470	9.393	13.842
2	0.0000	***	0.0045	**	0.0000	***	0.0001	***	20.092	11.190	1.470
4	0.0000	***	0.3121	**	0.0000	***	0.0000	***	0.510	0.290	11.830
5	0.0000	***	0.0000	***	0.0088	**	0.0000	***	0.560	12.903	1.550
7	0.0005	***	0.0000	***	0.0000	***	0.2781		0.330	5.250	4.500
8	0.0015	**	0.2189		0.0008	***	0.0316	*	2.660	1.910	0.700
9	0.0053	**	0.0000	***	0.0002	***	0.0027	**	0.410	3.310	1.610
10	0.0001	***	0.0011	**	0.0000	***	0.2324		6.881	1.020	0.741
13	0.0000	***	0.0432	*	0.0000	***	0.0136	*	1.370	1.870	3.030
17	0.0000	***	0.0001	***	0.0000	***	0.7726		5.620	0.220	0.180
18	0.0145	*	0.0780	.	0.0000	***	0.0030	**	2.290	1.560	0.550
19	0.0000	***	0.0044	**	0.0000	***	0.0000	***	0.060	0.360	2.930

Stage III:

Cluster= C	pvalOwa		p_MWU_P_S3 vs N_S3		p_MWU_P_S3 vs T_S3		p_MWU_N_S3 vs T_S3		Median_P_S3	Median_N_S3	Median_T_S3
0	0.0003	***	0.0770	.	0.0075	**	0.0003	***	8.280	5.360	16.690
1	0.0028	**	0.0001	***	0.0000	***	0.0092	**	1.200	10.927	3.710
2	0.0022	**	0.0770	.	0.0011	**	0.0353	*	19.600	11.520	3.810
4	0.0001	***	0.0770	.	0.0002	***	0.0000	***	1.800	0.780	8.900
5	0.0000	***	0.0000	***	0.0000	***	0.0000	***	0.580	12.420	3.193
7	0.0000	***	0.0004	***	0.0000	***	0.0482	*	0.320	2.660	6.536
8	0.0193	*	0.0142	*	0.0006	***	0.1276		6.540	3.620	2.400
9	0.0189	*	0.0005	***	0.0000	***	0.8973		0.400	3.143	3.050
10	0.0000	***	0.0400	*	0.0003	***	0.0630	.	8.380	1.640	0.831
13	0.0045	**	0.2508		0.0031	**	0.0649	.	1.060	1.260	2.680
17	0.0000	***	0.0004	***	0.0000	***	0.0726	.	10.160	0.560	0.260
18	0.0000	***	0.1903		0.0001	***	0.0015	**	2.128	1.960	0.510
19	0.0002	***	0.0315	*	0.0001	***	0.0208	*	0.300	0.700	1.510

Note: Statistical test by repeated measure one-way ANOVA test with paired-wise Mann-Whitney U test between groups.

New Suppl. Fig. 2:

(3) When comparing clusters across different disease stages, no common point of comparison was found across the different clusters, with S2 being compared to S1 in some clusters and to S3 in other clusters. The authors didn't address this point clearly, as only a specification was added in the text for the nature of each comparison performed (i.e. the control stage vs. S2). While the results of the manual gating of key immune subsets helps to partially address this problem, the previously presented results remain to be further adjusted, with the potential removal of some of the findings.

R: We examined the immune changes across tumour stages: I, II & III using one-way ANOVA test with Tukey post-hoc multiple comparison test to identify any cluster showing differences in an **unbiased manner**. Of course we then used the manual gating to validate these immune subsets with the defined immune phenotypes from the initial unbiased CyTOF analysis. We feel that it is necessary to present all data in the unbiased manner rather than removing some of them.

(4) For the scRNA-seq analysis, the authors were asked to specify the number of cells used for each samples/patient and each cluster, and to state whether each cluster consisted of cells from more of one patient. Supplementary Figure 3A shows the distribution of clusters in each patient. However, the inverse should have been evaluated (i.e. the contribution of patients to each cluster), hence helping to appreciate whether some clusters were driven by few patients. Additionally, the authors indicate that “each of the T, NK and MNPs clusters with total cell numbers as well as proportions contributed by each patient sample” are provided in Supplementary Table 5. However, Supplementary Table 5 displays the “Top 25 genes for T, NK and myeloid/DC clusters by scRNA sequencing analysis (adapted from Sharma et al. Cell 2020)”, and the specified numbers/proportions are not found in any of the Supplementary Tables. This should be adjusted accordingly.

R: To address this concern, we have revised and represented this data in reversed, by plotting proportion of each patient’s contribution towards each cluster (see the revised Suppl Fig. 3a below). As the **number of patient samples are more**, particularly as we performed multi-regional tumour (T) sequencing as specified in Methods, hence they have to be presented vertically. That is also why we hesitated to present the data this way and thought that the original presentation is clearer and could somewhat provide the information on distribution of clusters within each patient. However, we understand and agree to present the data to illustrate patient samples’ contribution to each cluster. From the data below, inevitably, some samples contributed more while others less to some clusters as the natural variation of the sample characteristics. However, overall, there is no obvious over-representation by any sample, particularly if segregated according to stages (**actual cell number per sample and per cluster as attached in a separate table**).

Revised Suppl Fig. 3a: patient samples are presented as each colour bar:

Second, we believe there might be misinterpretation of our data in Suppl. Table 5 (now Suppl Table 6 as we incorporated new data). The exact description in the revised manuscript was: “Pseudotime trajectory analysis of tumour-infiltrating immune cells were performed using

the Monocle R package (version 3.0)20, on T cells (further sub-divided to CD3+CD8- and CD3+CD8+ T cells), NK cells and MNPs clusters (Supplementary Fig. 4a and Supplementary Table 6).” In fact, Suppl Table 6 (original Suppl Table 5) shows the top 25 genes for each scRNA seq clusters while the **total number of cells** for each of the immune lineages (T, NK and MNPs) were represented in Suppl Fig. 4a. We have now provided the actual number of cells per pat sample per cluster as a **table separately attached**. However, we felt that this data might be too much and less relevant as the **most important and relevant data for our current study** is actually the **immune trajectory data** (Figure 3 & Suppl Fig.5 b), which focused on specific markers expressions. The clustering data was **adapted from Sharma et al. Cell 2020 paper**, which was already published by our co-author Prof R. DasGupta (<https://doi.org/10.1016/j.cell.2020.08.040>) and they were used merely as supporting data to demonstrate the various clusters from the scRNA seq data. Therefore, we believe that Revised Suppl Fig. 3a could sufficiently address the concerns on the contribution of each patient sample to each cluster and would suggest that it is not necessary to include the attached table with actual cell numbers in the manuscript.

(5) The definitions of clonal and subclonal mutations presented by the authors are not typical. Importantly, a mutation can be present in all biopsy sites (or tumor regions) and still be subclonal. The authors didn't address this issue appropriately and cited the paper by McGranahan N. et al. (Science, 2016) to justify their definition of subclonality and clonality. However, it is clearly stated in the same paper (referenced by the authors in their response) that clonal mutations are “present in all tumor cells” (as opposed to all biopsy sites of a tumor), while subclonal mutations are “present in only a subset [of cells]”. A simple and clear example of this would be a mutation that is present in only 10% of tumor cells, but is found in 10% of tumor cells in all biopsy sites / tumor regions. This is still, by definition, a SUBCLONAL mutation (found in only 10% of tumor cells), even though it might be seen in all tumor regions.

R: We can see why the definition can be rather misleading in the McGranahan et al Science 2016 paper (<https://science.sciencemag.org/content/351/6280/1463>). The description: “clonal (present in all tumor cells) versus subclonal (present only in a subset) neoantigens” were **estimated from TCGA cohort**, where only a single biopsy or tumour samples were sequenced. Hence, as quote from the article “*to determine clonality from sequencing of a single sample, the cancer cell fraction, which describes the proportion of cancer cells harboring a mutation, was determined for each neoantigen*”. This itself in fact is an **estimation** using an artificial ITH threshold cut off as illustrated in Fig. 1B from McGranahan et al Science 2016. Also since they **did not perform single cell sequencing** in their paper, it is actually **not likely to accurately assess data from “all tumour cells”**. Instead, in their own cohort, they performed **multi-region sequencing**, a method we also adopted in our HCC cohort as Prof Charlie Swanton was actually the science advisory board member who helped us to establish the sampling protocols for our HCC study at the initial phase: <https://clinicaltrials.gov/ct2/show/NCT03267641>. To further support our clonal and subclonal definitions, the description was actually provided in the **supplementary materials and methods** from the same paper: <https://science.sciencemag.org/content/sci/suppl/2016/03/02/science.aaf1490.DC1/McGranahan-SM.pdf>.

“Clonal architecture analysis

*For samples subject to **multi-region sequencing**, clonal status of each mutation was estimated based on multi-region sequencing calls. In brief, each mutation was*

*classified as **clonal if identified and present in each and every tumor region sequenced within the tumor.** Conversely, **any mutations not ubiquitously present in every tumor region were classified as subclonal.**”*

The same definition was also applied to our previously published papers: Nguyen et al Nat Com 2021 (<https://www.nature.com/articles/s41467-020-20171-7>) and Zhai et al Nat Com 2017. Therefore, we are confident that the definition of clonal and subclonal neoantigens are correct and is the most relevant definitions for our multiregional whole genome sequencing cohort.

(6) In their response, the authors argue that PD-1+ T-cells represent most likely an exhausted population as “these are TILs within TME, which are most likely chronically exposed tumor antigens”. This statement is not completely correct, as it has been previously shown that CD8+ PD-1+ tumor-infiltrating cells correlate with high levels of T-cell activation and are associated with better outcomes (Pignon J.C. et al., Clinical Cancer Research, 2018; PMID: 30670497). While the analysis of immune checkpoint markers (i.e. LAG-3, TIM-3) on PD-1+ cells favors the exhausted nature of these cells, the first statement should be clarified.

R: Indeed, we acknowledged and agreed with the reviewer on this idea, therefore we actually did not include this statement on “chronic exposure to tumour antigens” in the manuscript text but rather toned down the claim as much as possible. The hypothesis was suggested merely as a possible explanation for the exhaustion status of these cells, which is supported by the expression of multiple exhaustion markers on these cells (Suppl. Fig. 4c, 4d). We reassure that such statement does not appear in the manuscript text and as much as possible as underlined in the text we did not make strong claim that these cells are “exhausted” rather we included the word “potentially” exhausted.

(7) The authors were asked to externally validate their findings using publicly available datasets, such as the TCGA. The findings do not appear to validate in the TCGA. The authors note that this “may due mainly to a different background of these HCC patients in TCGA cohort, who may receive different prior therapies, such as radiotherapy and systemic therapy, which will change the immune landscapes”. I do not believe this is correct – the sample inclusion criteria for HCC for TCGA (from the original 2017 Cell manuscript) specifically states it is patients who had not received prior treatment for their disease. With this in mind, it is concerning that the findings do not appear to validate in an external dataset.

R: According to the TCGA cohort we downloaded from

<https://portal.gdc.cancer.gov/projects/TCGA-LIHC>

The clinical record of these patients does indicate multiple treatments including Radiation therapy and pharmaceutical therapy (see the excel file, below, in the last column) this was confusing for us at first. Although we do also note the statement regarding no prior treatment in the Cell paper.

case_id	case_submitter_id	project_id	treatment_or_therapy	treatment_outcome	treatment_type
03f5de7d-45a9-4a73-8c61-ed816d4a3ac5	TCGA-DD-AAD2	TCGA-LIHC	no	'--	Radiation Therapy, NOS
03f5de7d-45a9-4a73-8c61-ed816d4a3ac5	TCGA-DD-AAD2	TCGA-LIHC	no	'--	Pharmaceutical Therapy, NOS
91a0c8cd-4f23-4569-8e5c-e5442f7af530	TCGA-4R-AA8I	TCGA-LIHC	no	'--	Radiation Therapy, NOS
91a0c8cd-4f23-4569-8e5c-e5442f7af530	TCGA-4R-AA8I	TCGA-LIHC	no	'--	Pharmaceutical Therapy, NOS
5c08008d-5ed4-4480-bd00-c8a2d8c787dc	TCGA-FV-A4ZP	TCGA-LIHC	no	'--	Radiation Therapy, NOS
5c08008d-5ed4-4480-bd00-c8a2d8c787dc	TCGA-FV-A4ZP	TCGA-LIHC	no	'--	Pharmaceutical Therapy, NOS
4c960eee-e4b5-499d-a596-238aa78745a4	TCGA-DD-AAE9	TCGA-LIHC	no	'--	Pharmaceutical Therapy, NOS
4c960eee-e4b5-499d-a596-238aa78745a4	TCGA-DD-AAE9	TCGA-LIHC	no	'--	Radiation Therapy, NOS
08428220-abd4-484e-85ca-3673454c837e	TCGA-G3-A25X	TCGA-LIHC	no	'--	Radiation Therapy, NOS
08428220-abd4-484e-85ca-3673454c837e	TCGA-G3-A25X	TCGA-LIHC	no	'--	Pharmaceutical Therapy, NOS
1ad7f17e-b3b8-40e9-aab8-5a17e0aec408	TCGA-CC-5261	TCGA-LIHC	no	'--	Pharmaceutical Therapy, NOS
1ad7f17e-b3b8-40e9-aab8-5a17e0aec408	TCGA-CC-5261	TCGA-LIHC	no	'--	Radiation Therapy, NOS
3528dd91-1a6d-4d41-a3ee-829c857cc904	TCGA-2Y-A9H8	TCGA-LIHC	yes	'--	Pharmaceutical Therapy, NOS
3528dd91-1a6d-4d41-a3ee-829c857cc904	TCGA-2Y-A9H8	TCGA-LIHC	no	'--	Radiation Therapy, NOS
1f482810-433a-4d46-b81b-f7909fff8c09	TCGA-DD-AAD1	TCGA-LIHC	no	'--	Pharmaceutical Therapy, NOS
1f482810-433a-4d46-b81b-f7909fff8c09	TCGA-DD-AAD1	TCGA-LIHC	no	'--	Radiation Therapy, NOS
8574d285-e02a-4d2b-931b-73556a9fda02	TCGA-MR-A8JO	TCGA-LIHC	no	'--	Radiation Therapy, NOS
8574d285-e02a-4d2b-931b-73556a9fda02	TCGA-MR-A8JO	TCGA-LIHC	no	'--	Pharmaceutical Therapy, NOS
c52abac9-07da-4b3d-bf8a-2161cacf9682	TCGA-ED-A4XI	TCGA-LIHC	no	'--	Pharmaceutical Therapy, NOS
c52abac9-07da-4b3d-bf8a-2161cacf9682	TCGA-ED-A4XI	TCGA-LIHC	no	'--	Radiation Therapy, NOS
0d0a10b4-7b99-4043-a129-89844f258156	TCGA-DD-A39X	TCGA-LIHC	no	'--	Pharmaceutical Therapy, NOS
0d0a10b4-7b99-4043-a129-89844f258156	TCGA-DD-A39X	TCGA-LIHC	no	'--	Radiation Therapy, NOS
290890eb-321c-45dc-9cd1-45676c6e5e96	TCGA-5R-AA1C	TCGA-LIHC	no	'--	Pharmaceutical Therapy, NOS
290890eb-321c-45dc-9cd1-45676c6e5e96	TCGA-5R-AA1C	TCGA-LIHC	no	'--	Radiation Therapy, NOS
609c82fd-eb3a-46d7-8c09-0cde281a50b3	TCGA-DD-A11A	TCGA-LIHC	not reported	'--	Pharmaceutical Therapy, NOS
609c82fd-eb3a-46d7-8c09-0cde281a50b3	TCGA-DD-A11A	TCGA-LIHC	not reported	'--	Radiation Therapy, NOS
97472e71-66eb-434b-81b0-d727765f44cb	TCGA-DD-AADI	TCGA-LIHC	no	'--	Pharmaceutical Therapy, NOS
97472e71-66eb-434b-81b0-d727765f44cb	TCGA-DD-AADI	TCGA-LIHC	no	'--	Radiation Therapy, NOS
cf358fb8-866c-4780-a550-d6c1c30b8c63	TCGA-DD-AADU	TCGA-LIHC	no	'--	Pharmaceutical Therapy, NOS
cf358fb8-866c-4780-a550-d6c1c30b8c63	TCGA-DD-AADU	TCGA-LIHC	no	'--	Radiation Therapy, NOS
23103931-cc29-4f89-ae9c-79d53401ecea	TCGA-BC-A10U	TCGA-LIHC	no	'--	Pharmaceutical Therapy, NOS
23103931-cc29-4f89-ae9c-79d53401ecea	TCGA-BC-A10U	TCGA-LIHC	no	'--	Radiation Therapy, NOS
714a6878-ed2a-47f3-9755-b705ac144d68	TCGA-RC-A7SF	TCGA-LIHC	no	'--	Radiation Therapy, NOS
714a6878-ed2a-47f3-9755-b705ac144d68	TCGA-RC-A7SF	TCGA-LIHC	no	'--	Pharmaceutical Therapy, NOS
afcb9708-92b3-4727-9c0d-63e7ff22d87b	TCGA-DD-AAVX	TCGA-LIHC	yes	'--	Pharmaceutical Therapy, NOS
afcb9708-92b3-4727-9c0d-63e7ff22d87b	TCGA-DD-AAVX	TCGA-LIHC	yes	'--	Radiation Therapy, NOS

Despite that, from the TCGA data, as also provided in our previous rebuttal letter, we did in fact observe a progressive downtrend of multiple key genes (Two-way Anova $p < 0.05$) involved in antigen presentation, inflammatory response, exhaustion and CD8 T chemotaxis validating the key immune processes downregulated along tumour progression (see figure below), despite a more modest differences among stages. A possible reason for this small discrepancy is because our data is based on the averaged data from multiregional tumour sampling (rather than a single biopsy data from TCGA cohort), which given the known intratumoural heterogeneity of HCC (Nguyen et al Nat Com 2021 and Zhai et al Nat Com 2017), would provide a more accurate data. We have now included this TCGA validation data in the **new suppl Fig. 6b** as additional support for our data.

New Suppl. Fig. 6b: TCGA interrogation of key genes in manuscript:

Antigen presentation

p = 0.0014 **

Inflammatory response

p = 0.0162 *

Exhaustion

p = 0.0110 *

CD8 T chemotaxis

p < 0.0001 ****

REVIEWER COMMENTS

Reviewer #1 (Remarks to the Author):

There is no additional point to address for the review on my part. Although this study still holds the considerable limitations in the size of human sample and the discrepancy between the mouse model and the clinical cases, the author had reached out the reasonable conclusions. The data in this study open to the public would be useful to further study for HCC.

Reviewer #2 (Remarks to the Author):

After this round of revision, my comments are the same:

-Impressive huge amount of data

-Without external validation, there is a major risk of an over-interpretation of the main conclusion of the paper that could be related to the selection of the S3 group of patients as it was reported in R1 rebuttal by the authors

Reviewer #3 (Remarks to the Author):

The authors have addressed many of our prior comments. The following outstanding issues remain:

Comment (1): (adjustment of CyTOF analysis to understand if clusters are patient-specific)

- The authors added a heatmap trying to address this point, however, in the heatmap designed by the authors in New Supp. Fig. 1a, the authors show the composition of each sample in terms of clusters (i.e. the color scale corresponds to the percentage of clusters within each sample, and not to the percentage of each sample within clusters). This doesn't illustrate the composition of each cluster in terms of samples and consequently doesn't help to tell appropriately if a cluster is patient-specific or not. This could be corrected by inverting the color scale so that it corresponds to percentage of samples with each cluster.

Comment (2): (quantifying immune diversity in tumor samples)

- While the authors addressed this comment through a well-defined approach and assessed quantitatively the immune diversity, I am not familiar with the metric used (i.e. the multivariate beta-dispersion of the Bray-Curtis distances between immune clusters for each tissue type). I would suggest review of this method by a statistician.

Comment (5): (concerning the definitions of clonal and subclonal events adopted by the authors)

- I again respectfully disagree with the authors. As a simple example, if a mutation is present in only 10% of tumor cells and not all tumor cells, it is, by definition, a subclonal event. Now if it is present in only 10% of tumor cells when multiple regions are sample, but is still NOT found in 90% of tumor cells in every region sampled, it is still, by definition, a subclonal event. While single-cell sequencing was

not performed, the fraction of tumor cells harboring a mutation can be inferred using a number of available tools (they can calculate a cancer cell fraction, which should be 1 for a clonal mutation)

For comment (7): (regarding validation using the TCGA data)

- Two small points here. First, I would ask the authors double check the statement about TCGA samples receiving treatment PRIOR to specimen collection (my understanding is that the treatment annotation is AFTER the samples were collected, and so any subsequent treatment would not affect the analyzed tumor itself). Second, I would ask the authors ensure they are using the same statistical methods in their analysis of TCGA data as in the analysis of their own primary data (e.g. if they used pairwise testing they should continue to do so, if they used ANOVA previously then continue to do so, etc.).

Point-by-point response to reviewers' comments marked with “R”:

Reviewer #1 (Remarks to the Author):

There is no additional point to address for the review on my part. Although this study still holds the considerable limitations in the size of human sample and the discrepancy between the mouse model and the clinical cases, the author had reached out the reasonable conclusions. The data in this study open to the public would be useful to further study for HCC.

R: We thank this reviewer for the comments. We acknowledge the limitation of the current study and will work towards collecting more samples in the future.

Reviewer #2 (Remarks to the Author):

After this round of revision, my comments are the same:

-Impressive huge amount of data

-Without external validation, there is a major risk of an over-interpretation of the main conclusion of the paper that could be related to the selection of the S3 group of patients as it was reported in R1 rebuttal by the authors

R: Once again we thank the reviewer for the comments and we fully acknowledge the limitation of our data. Even though, we did provide further validation of our main findings in an independent single-cell RNA seq trajectory analysis (n=14 patients, **Fig. 3**), another independent FFPE tissue-microarray (TMA) cohort (n= 102 patients, **Fig. 5e** and **5f**), TCGA cohort (n= 297 patients, **suppl. Fig. 6b**) as well as a HCC murine model (**Fig. 6**). Despite that, we do acknowledge and agree with the fact that in clinical practice more advanced patients will definitely be more carefully considered for resection. Given this point, we have chosen to **tone down all our claims specific to Stage 3** as “potential” immune recovery. We also agree that this is an important observation that needs further validation in future and hence have added the following statements in Discussion (as underlined):

“However, given the fact that most advanced HCC patients may not undergo resection as first-line therapy, this interesting finding will require careful interpretation against this inevitable confounding factor.”

We hope this will help readers fully understand the limitation of our study when interpreting our findings.

Reviewer #3 (Remarks to the Author):

The authors have addressed many of our prior comments. The following outstanding issues remain:

*Comment (1): (adjustment of CyTOF analysis to understand if clusters are patient-specific)
- The authors added a heatmap trying to address this point, however, in the heatmap designed by the authors in New Supp. Fig. 1a, the authors show the composition of each sample in terms of clusters (i.e. the color scale corresponds to the percentage of clusters within each sample, and not to the percentage of each sample within clusters). This doesn't illustrate the composition of each cluster in terms of samples and consequently doesn't help*

to tell appropriately if a cluster is patient-specific or not. This could be corrected by inverting the color scale so that it corresponds to percentage of samples with each cluster.

R: Thank you for the suggestion. We have plotted the heatmap inverted by showing the proportions of samples within each cluster (**revised suppl Fig. 1a** and below). As shown, most clusters have relatively even contributions by the samples, except for smaller clusters (<0.5% e.g. cluster 31 and 33) which could be unevenly represented by the samples. Most importantly, the clusters that we identified as significantly different across tumours from Stage 1-3 HCC: i.e. C1, C2, C4, C6, C9 and C19 (**Fig. 2d**) showed relatively equivalent sample distributions.

Revised suppl Fig. 1a:

Comment (2): (quantifying immune diversity in tumor samples)

- While the authors addressed this comment through a well-defined approach and assessed quantitatively the immune diversity, I am not familiar with the metric used (i.e. the multivariate beta-dispersion of the Bray-Curtis distances between immune clusters for each tissue type). I would suggest review of this method by a statistician.

R: We understand this may be a new and unfamiliar approach. This data is actually generated by our co-first author, Dr Martin Wasser, who is a bioinformatics scientist and principal biostatistician in our institution with 20 years' experience.

To provide further information, multivariate beta-dispersion of the Bray-Curtis distances or, in short, Bray-Curtis dissimilarity is a well-established method in ecology to compare species diversity (1). Our and other labs have also used Bray-Curtis dissimilarity in microbiome studies (2). For dissimilarity of our CyTOF data, we first determined the median centroids of all groups (P, N, T) based on the percentages of the clusters. Subsequently, distances/dissimilarities of all samples to their respective centroid are calculated based on the Bray-Curtis distance metric that ranges between 0 (minimum) to 1 (maximum dissimilarity). For the implementation, we used the well-known vegan (version 2.5.7) R package (3).

References:

1. Bray, J. R. & Curtis, J. T. An Ordination of the Upland Forest Communities of Southern Wisconsin. *Ecological Monographs* **27**, 325–349 (1957).
2. Mishra, A. *et al.* Microbial exposure during early human development primes fetal immune cells. *Cell* **184**, 3394–3409.e20 (2021).

3. Oksanen, J. et al. vegan: Community Ecology Package. (2020). <https://CRAN.R-project.org/package=vegan>

Comment (5): (concerning the definitions of clonal and subclonal events adopted by the authors)

- I again respectfully disagree with the authors. As a simple example, if a mutation is present in only 10% of tumor cells and not all tumor cells, it is, by definition, a subclonal event. Now if it is present in only 10% of tumor cells when multiple regions are sample, but is still NOT found in 90% of tumor cells in every region sampled, it is still, by definition, a subclonal event. While single-cell sequencing was not performed, the fraction of tumor cells harboring a mutation can be inferred using a number of available tools (they can calculate a cancer cell fraction, which should be 1 for a clonal mutation)

R: We totally agree with the reviewer that % mutation could help in estimating clonal or subclonal mutation within one particular tumour sector, commonly used when a single tumour was assayed. We have adopted a “presence” or “absence” approach with multi-regional tumour sampling where we defined neoantigens as expressed by all tumour sectors/regions or not following the methods from McGranahan et al Science 2016 detailed in supplementary materials and methods

<https://science.sciencemag.org/content/sci/suppl/2016/03/02/science.aaf1490.DC1/McGranahan-SM.pdf>. We however agree with the reviewer that this shows more of the “spatial”

intratumoural heterogeneity and that we could not assume clonality in each region. Even though the significant link to stages we observed is with the subclonal neoantigens, which will remain true as they were absent in some tumour sectors based on our definition. Hence we would like to propose to use the terms **ubiquitous** and **heterogenous** neoantigens instead (changes underlined in revised manuscript and shown below), the same definition was used to evaluate the intratumour heterogeneity of nonsilent mutations, according to regions instead of all tumour cells, in Swanton’s paper in Science 2014

<https://www.science.org/doi/10.1126/science.1253462>

“We quantified neoantigens in tumour samples from each stage using whole-genome sequencing (Methods) and found significantly more heterogenous (occurring in at least one but not all tumour sectors) but not ubiquitous (occurring in all tumour sectors) neoantigens in S3 than S1 tumours (**Fig. 4f**).”

For comment (7): (regarding validation using the TCGA data)

- Two small points here. First, I would ask the authors double check the statement about TCGA samples receiving treatment PRIOR to specimen collection (my understanding is that the treatment annotation is AFTER the samples were collected, and so any subsequent treatment would not affect the analyzed tumor itself). Second, I would ask the authors ensure they are using the same statistical methods in their analysis of TCGA data as in the analysis of their own primary data (e.g. if they used pairwise testing they should continue to do so, if they used ANOVA previously then continue to do so, etc.).

R: To clarify these points, we have managed to download the full clinical data for the TCGA cohort including information on prior treatment (**see the screenshot shown below**). Hence we would like to clarify that yes indeed, most of the TCGA data is without prior treatment and from the data we downloaded from the website (n=299), we have only two patients who have received prior treatment. Therefore, we repeated our analysis by filtering off these two samples from patients who received prior treatment and updated them in Suppl Fig. 6b. As

expected it hardly changed the p values at all, except for one of two cases with marginal changes observed (e.g. p values from 0.0014 to 0.0013). Therefore, our conclusion remains robust.

We however do not find further information/explanation on the types of treatment indicated in the clinical info (as also shown in the screenshot shown below) but we believe these could likely be post-surgical treatment types. In the Cell paper, we believe that the authors have filtered the data or selected the data containing only patients without prior treatment as stated in their paper "**196 cases from LIHC batches 100, 131, 153, 173, 203, 231, 275, 287, 303, 314, 327, 341, 345, and 365**" (quote from their method).

[https://www.cell.com/cell/fulltext/S0092-8674\(17\)30639-6](https://www.cell.com/cell/fulltext/S0092-8674(17)30639-6).

	case_id	case_submitter_id	project_id	prior_malignancy	prior_treatment	treatment_or_therapy	treatment_type
5	948a9bbc-60b0-4fb7-a707-76bc41294217	TCGA-LG-A9QD	TCGA-LIHC	no	No	yes	Pharmaceutical Therapy, NOS
6	2bea076b-c923-4a91-a37b-c6615fec021c	TCGA-ED-A8O5	TCGA-LIHC	no	No	no	Pharmaceutical Therapy, NOS
9	470d52b8-3d74-4789-b714-b9869a96657b	TCGA-ED-A7PX	TCGA-LIHC	no	No	no	Radiation Therapy, NOS
10	4db9de75-9297-46e5-acc5-9275f52a642a	TCGA-XR-A8TC	TCGA-LIHC	yes	Yes	no	Pharmaceutical Therapy, NOS
12	80e7742f-abbe-4f11-b4e4-c999dc1bd459	TCGA-DD-AAVQ	TCGA-LIHC	no	No	no	Pharmaceutical Therapy, NOS
14	fb0e4bc-5398-4a7e-963c-86ea1108a2de	TCGA-DD-AAEH	TCGA-LIHC	no	No	no	Pharmaceutical Therapy, NOS
16	3837c8ac-6c9e-499b-a4e7-d42c870bcb49	TCGA-CC-A5UD	TCGA-LIHC	no	No	no	Radiation Therapy, NOS

To the second point, yes we did use **consistent statistical tests for the analysis** in our own cohort and TCGA cohort analysis. For comparison of multiple genes representing the same pathways across different stages of HCC, we used Two-way Anova (used to analyze the difference between the means of more than two groups) followed by respective p values from Tukey's comparison test between different stages. As patients from different stages are different patients the test is unpaired. All these are already **specified in the figure legends** for both **Fig. 4d & e** and **Fig. 5d** (our dataset) and **suppl. Fig. 6b** (TCGA validation data).

REVIEWER COMMENTS

Reviewer #3 (Remarks to the Author):

The authors have addressed all of the reviewer comments.